



# Friagem Event in Central Amazon and its Influence on Micrometeorological Variables and Atmospheric Chemistry

Guilherme F. Camarinha-Neto[1], Julia C. P. Cohen[1,2], Cléo Q. Dias-Júnior[3], Matthias Sörgel[4,a], José Henrique Cattanio[1,2], Alessandro Araújo[5], Stefan Wolff[4,b], Paulo A. F. Kuhn[1], Rodrigo A. F. Souza[6], Luciana V. Rizzo[7], and Paulo Artaxo[8]

[1]Federal University of Pará (UFPA), Postgraduate Program on Environmental Sciences - PPGCA, Belém, AM, Brazil
[2]Faculty of Meteorology, Federal University of Pará (UFPA), Belém, PA, Brazil
[3]Department of Physics, Federal Institute of Pará (IFPA), Belém, PA, Brazil
[4]Biogeochemistry Department, Max Planck Institute for Chemistry, P.O. Box 3060, 55020 Mainz, Germany
[5]Empresa Brasileira de Pesquisa Agropecuária (EMBRAPA), Belém, PA, Brazil
[6]Department of Meteorology, Amazonas State University (UEA), Manaus, Amazonas, Brazil
[7]Department of Environmental Sciences, Institute of Environmental, Chemical and Pharmaceutics Sciences, Universidade Federal de São Paulo (UNIFESP), São Paulo, São Paulo, Brazil
[8]Institute of Physics, University of São Paulo (USP), São Paulo, São Paulo, Brazil
[a]currently at: Atmospheric Chemistry Department, Max Planck Institute for Chemistry, P.O. Box 3060, 55020 Mainz, Germany
[b]currently at: Multiphase Chemistry Department, Max Planck Institute for Chemistry, P.O. Box 3060, 55020 Mainz, Germany

**Correspondence:** Cléo Q. Dias-Júnior (cleo.quaresma@ifpa.edu.br)

**Abstract.** In the period between July $9^{th}$ and $11^{th}$, 2014 a Friagem event reached the central Amazon region causing significant changes in microclimate and atmospheric chemistry. On July $11^{th}$, the southwest flow related to the Friagem converged with the easterly winds in the central Amazon region. The interaction between these two distinct air masses formed a convection band, which intensified over the Manaus region and the Amazon Tall Tower Observatory (ATTO) site. The satellite images

show the evolution of convective activity on July $11^{th}$, which lead to $21\ mm$ of precipitation in the ATTO site. Moreover, the arrival of the Friagem caused a sudden drop in temperature and a predominance of southerly winds, which could be seen in Porto Velho between July $7^{th}$ and $8^{th}$ and in Manaus and ATTO site from July $9^{th}$ to $11^{th}$. The results of ERA-Interim reanalysis and Brazilian developments on the Regional Atmospheric Modeling System (BRAMS) simulations show that this Friagem event coming from the southwest, carries a mass of air with higher $O_3$ and $NO_2$ mixing ratios and lower CO mixing

ratio compared to the airmasses present at the central Amazon. At lake Balbina the Friagem intensifies the local circulations, such as the breeze phenomena. At the Manaus region and ATTO site, the main effects of the Friagem event are: a decrease in the incoming solar radiation (due to intense cloud formation), a large temperature drop and a distinct change in surface $O_3$ and $CO_2$ mixing ratios. As the cold air of the Friagem was just in the lower $500\ m$ the most probable cause of this change is that a cold pool above the forest prevented vertical mixing causing accumulation of $CO_2$ from respiration and very low $O_3$ mixing

ratio due to photochemistry reduction and limited mixing within the boundary layer.





## 1 Introduction

The Amazon region suffers from the incursion of cold waves from the high latitudes of the Southern hemisphere (SH), with a relatively common occurrence mainly in the less rainy season, between June and September. These events are denominated
locally and in literature as Friagem and about $70\%$ of the cases of Friagem occur in this period of the year (Brinkmann and Ribeiro, 1972; Marengo et al., 1997; Fisch et al., 1998; de Oliveira et al., 2004; Caraballo et al., 2014). Brinkmann and Ribeiro (1972) observed 2 to 3 Friagem events per year, preferably in the less rainy season, in the central Amazon. This was one of the first studies to explore frontal system (FS) interference in central Amazon.

Silva Dias et al. (2004) showed that the arrival of a Friagem event in the West of the Amazon generates a pressure gradient
force whose direction is opposite to the trade winds, thus causing a weakening of these winds. These authors observed that the weakening of the trade winds enables the development of vigorous local circulations in the region of Santarém - PA.

Moura et al. (2004), who used data collected at the shores of Lake Balbina (central Amazon), concluded that without the influence of large-scale flow it is possible to observe the dynamics of breeze circulations influencing the ozone ($O_3$) mixing ratio with more clarity. According to these authors, the $O_3$-mixing ratio changes are larger when the flow occurs in the direction
from the lake to the forest, that is, during the occurrence of the lake breeze.

In North America, Sun et al. (1998) evinced that when large-scale wind is weak, breeze circulations in Candle Lake, Canada, are more efficient in redistribution of heat, humidity, carbon dioxide ($CO_2$) and $O_3$ in the atmosphere. An important factor determining $O_3$ mixing ratio in a given region is the local variation in the wind field (Cheng, 2002). Therefore, the transport and dispersion of these trace gases are strongly affected by local wind systems, such as the breeze (Moura et al., 2004).

Marengo et al. (1997) compared the effects of the Friagem at Manaus (central Amazon) and Ji-Parana (south of the Amazon River), that are around $1,200\ km$ apart. They observed that the Friagem was strongly modified during its passage over the Amazon basin. For example, the lower temperatures in Ji-Parana could be associate to cold air advection, whereas in Manaus they were mainly caused by reduced solar radiation due to increased cloudiness.

Several studies have already shown the effect of the Friagem on the surface meteorological components (Marengo et al.,
1997; Fisch et al., 1998; Moura et al., 2004; Silva Dias et al., 2004). However, we are not aware of any study investigating the accompanied changes in trace gas concentrations and atmospheric chemistry in the Amazon Basin. Besides that, it is know that the presence of the Friagem phenomenon can alter the conditions of the local microclimate, allows the opportunity to better understand the dynamics of local circulations pattern and, consequently, influence local measurements carried out in Amazonian ecosystems, since they also cause the weakening of the predominant large-scale (trade) winds blowing from the
East in the study region (Silva Dias et al., 2004).

Therefore, the objective of this study is to investigate the effects of Friagem on micrometeorological variables measured in the Manaus region and in the forest region of the Amazon Tall Tower Observatory (ATTO) site (Andreae et al., 2015), as well as to evaluate the influence of this phenomenon on the local circulation dynamics and its role in the dispersion of trace gases at



ATTO site and Balbina lake. In order to achieve these objectives, the data measured in different sites around Manaus city and
ATTO-site and high resolution numerical simulation with the JULES-CCATT-BRAMS coupled model (Freitas et al., 2009)
were used.

## 2  Data and Methodology

### 2.1  Study area

The Sustainable Development Reserve (SDR) of Uatumã, São Sebastião do Uatumã county where the ATTO site is located
($02° 08' 38'' S$ - $59° 00' 07'' W$) is about $140\ km$ northeast of Manaus in the state of Amazonas, Brazil. The village of Balbina,
in Presidente Figueiredo county as well as the Balbina dam lake ($01° 52'\ S$ - $59° 30'\ W$), are located to the northwest of the
ATTO site (Fig. 1). The ATTO site is structured in a dense terra firme forest, where plateaus prevail, with a maximum elevation
of $138\ m$ (Andreae et al., 2015). The artificial lake of Balbina is a flooded area of approximately $1,700\ km^2$, with an average
depth of $10\ m$ (Kemenes et al., 2007).

Additionally, near surface measurements of $O_3$ made at T2 ($03.1392° S$ - $60.1315° W$), T3 ($03.2133° S$ - $60.5987° W$)
and the forest site T0z ($02.6091° S$ - $60.2093° W$) experimental sites, at a distance of 8, 70, and $60\ km$ from Manaus, re-
spectively, were used (Fig. 1). These sites were deployed in the Observations and Modeling of the Green Ocean Amazon
(GoAmazon2014/5) experiment (Martin et al., 2016). Due to its location, site T2 is heavily impacted by the Manaus urban
plume as well as emissions from brick factories and to a minor extent by local pollution sources such as shipping or burning
of household waste and wood near the site (Martin et al., 2016). The site T3 is typically downwind of Manaus city, influenced
by urban air masses in $38.5\%$ of time (Thalman et al., 2017). The site T0z, typically upwind Manaus (Rizzo et al., 2013), is
situated in the Cuieiras Biological Reserve ("ZF2") that has been a central part of Amazonian ecology and climate studies
for over 20 years (Araújo et al., 2002). These five sites will enable us to better understand the role of Friagem at near surface
$O_3$-levels in different parts of the central Amazon, some of them in the Manaus pollution plume (Cirino et al., 2018).

### 2.2  Data

A Friagem event that occurred between July $9^{th}$ and July $11^{th}$, 2014 in the region of the ATTO experimental site was iden-
tified and used as a case study. For this the data were collected at the ATTO site and at the international airports of Manaus
($03° 02' 08'' S$ - $60° 02' 47'' W$) and Porto Velho ($08° 42' 50'' S$ - $63° 53' 54'' W$), for July 2014. Air temperature data, as
well as wind direction and wind speed, in 30 min intervals were obtained from airport weather stations. The cities of Porto
Velho (about $930\ km$ southwest of ATTO) and Manaus (about $150\ km$ southwest of ATTO) were chosen with the purpose of
evaluating the impacts of the advance of the Friagem towards the region of the ATTO site.

The ATTO site air temperature, wind speed, wind direction, incident short-wave radiation and precipitation were measured at
the $81\ m$ high walk up tower ($02° 08.6470'\ S$ - $58° 59.9920'\ W$) at different heights (see table 1). $CO_2$ and $O_3$ measurements
were taken at 81 and $79\ m$ above ground, respectively. The measurements of $CO_2$ and $O_3$ mixing ratios were conceived



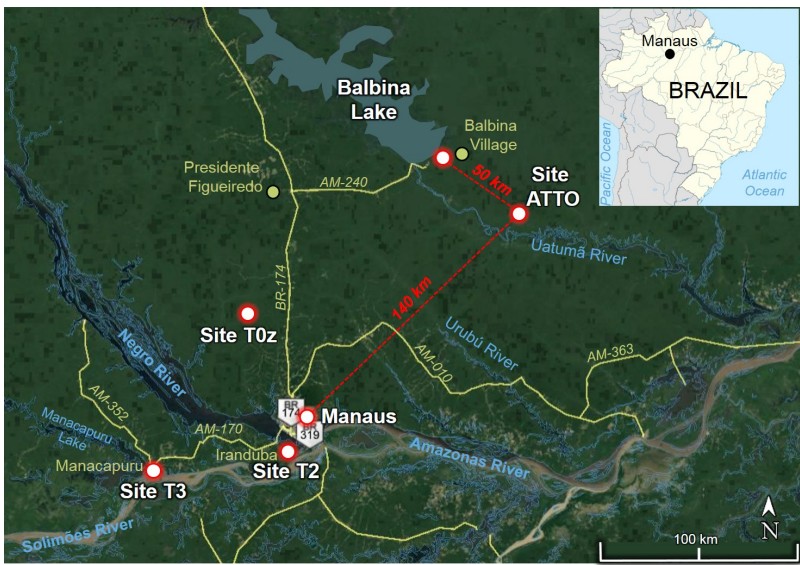

**Figure 1.** Google Earth map of the location of the ATTO site, Balbina lake, T2, T3 and T0z (white circles). The dashed red line indicates the distance from the ATTO site in relation to the Balbina lake and the city of Manaus (copyright: © Google Maps). The yellow lines represent the roads and the blue lines represent the network of the rivers in this region.

respectively by an infrared gas analyzer (IRGA, LI-7500A model, LI-COR inc., USA) and (TEI 49i model, Thermo Electron Corp, USA).

The data acquisition at the tower was performed by data loggers CR1000 and CR3000 (Campbell Scientific inc., USA), with instantaneous measurements taken every minute for meteorological variables and at high frequency for $CO_2$ (10 $Hz$) and $O_3$ (30 $s$) mixing ratio, subsequently processed every 30 min. The variables used in this study and their respective sensors are

presented in more detail in Table 1.

The $O_3$ data at T3 site were obtained as part of the U.S. Department of Energy Atmospheric Radiation Measurement Program (ARM, http://www.arm.gov/measurements) during the GoAmazon 2014/5 project (Martin et al., 2016). $O_3$-mixing ratios were measured with an ultra violet gas analyzer (TEI 49i model, Thermo Electron Corp, USA). The instrument was installed at a height of 3.5 $m$ above the ground (Dias-Júnior et al., 2017). At T2 and T0z, $O_3$-mixing ratios were also measured with the

same analyzer model (Thermo 49i) at a height of 12 $m$ a.g.l. and 39 $m$ a.g.l., respectively.

The European Center for Medium-Range Weather Forecasts (ECMWF) ERA-Interim reanalysis was used at intervals of 6 h, with the objective of evaluating the evolution of the Friagem event investigated in this work. The ERA-interim model and the ECMWF reanalysis system present spatial resolution with 60 vertical levels, harmonic spherical representation for the basic dynamic fields, and reduced Gaussian grid with uniform spacing of approximately 79 $km$ for the surface (Berrisford

et al., 2011). Furthermore, enhanced images of the infrared channel of the GOES-13 satellite were used, with the purpose of analyzing the formation and passage of convective systems in the study area.



**Table 1.** Variables used in this study, their respective measuring instruments and height in the micrometeorological tower at ATTO site.

| VARIABLES | INSTRUMENTS | HEIGHT |
|---|---|---|
| Air Temperature | **Thermo-hygrometer** (CS215, Campbell Scientific, USA) | 81 m |
| Wind Speed and Direction | **2D Sonic Anemometer** (Windsonic, Gill Instruments Ltd., UK) | 73 m |
| Incident Short Wave Radiation | **Pyranometer** (CMP21, Kipp and Zone, Netherlands) | 75 m |
| Rainfall | **Pluviometer** (TB4, Hydrological Services Pty. Ltd., Australia) | 81 m |
| $CO_2$-mixing ratio | **Infrared Gas Analyzer** (IRGA, LI-7500/LI-7200, LI-COR inc., USA) | 81 m |
| $O_3$-mixing ratio | **Ultraviolet Gas Analyzer** (TEI 49i, Thermo Electron Corp, USA) | 79 m |

## 2.3 Experimental design

The numerical simulations of the present study were made using the BRAMS (Brazilian Regional Atmospheric Modeling System) mesoscale model version 5.3 (Freitas et al., 2017). BRAMS represents a Brazilian version of the Regional Atmospheric 100 Modeling System (RAMS) (Cotton et al., 2003) adapted to tropical conditions. This version of BRAMS contains the coupling of the JULES (Joint UK Land Environment Simulator) (Best et al., 2011; Clark et al., 2011) and CCATT (Coupled Chemistry-Aerosol-Tracer Transport) models (Longo et al., 2010; Freitas et al., 2009), making BRAMS a new and fully-coupled numerical system of atmosphere-biosphere-chemical modeling, called JULES-CCATT-BRAMS (Moreira et al., 2013).

The integration time of the model was 72 hours, starting at 00 UTC on July $9^{th}$, 2014. The numerical experiment was 105 performed using only a grid whose horizontal resolution was $1.5 \ km$, with 185 points on $x$, 140 points on $y$, and 39 points on $z$. The vertical grid resolution was variable with the initial vertical spacing of $50 \ m$, increasing by a factor of 1.1 up to the $1.2 \ km$ level, and from that point forward this spacing was constant to the top of the model (around $16 \ km$). The domain covered by this grid, the distribution of the main rivers and topography can be observed in Fig. 2.

The initialization of the model was heterogeneous, using the ECMWF- ERA Interim reanalyses (www.ecmwf.int/en/forecasts/ 110 datasets/reanalysis) every 6 hours in a quarter-degree spatial resolution. Seven soil layers were defined up to the depth of $12.25 \ m$ and the assumed soil humidity was heterogeneous, as described in Freitas and Freitas (2006). Soil texture data were originally obtained from the Food and Agriculture Organization of the United Nations (UN FAO) and were adapted for the Brazilian territory by INPE (Rossato et al., 2004).

In this simulation, cloud microphysics uses the Thompson cloud water single-moment formulation, which consists of the 115 separate treatment of five classes of water that are then mixed in a single treatment for each type of cloud (Thompson et al.,





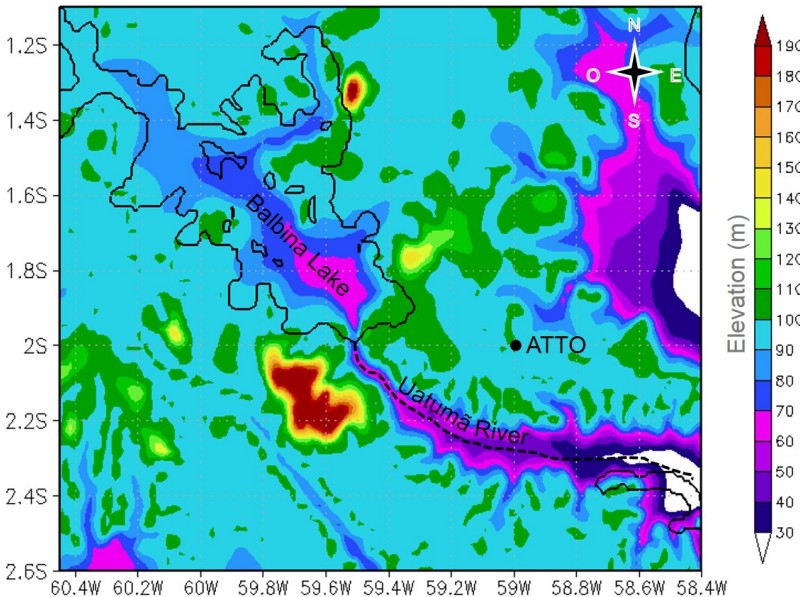

**Figure 2.** JULES-CCATT-BRAMS simulation grid showing the distribution of the topography ($m$) and location of the Balbina lake (black line), ATTO site (black point) and Uatumã river (dashed line)

2008; Thompson and Eidhammer, 2014). In addition, it includes the activation of aerosols in the cloud condensation nuclei (CCN) and ice nuclei (IN), thus, it predicts the concentration of the number of water droplets in the clouds, as well as the concentrations of two new aerosol variables, one for CCN and one for IN. These variables are grouped into hygroscopic aerosols called "water friendly" and non-hygroscopic aerosols are "ice friendly" (Freitas et al., 2017).

120  The parameterization of the long and short wave radiation used was the Carma (Community Aerosol and Radiation Model for Atmospheres) (Toon et al., 1989). This scheme solves the radiative transfer using the two-flux method and includes the main molecular absorbers (water vapor, $CO_2$, $O_3$ and $O_2$) and treats the gas absorption coefficients using an exponential sum formula (Toon et al., 1989). The JULES-CCATT-BRAMS radiation schemes are coupled online with the cloud and aerosol microphysics models to provide simulations of aerosol-cloud-radiation interactions (Freitas et al., 2017). The physical and

optical properties of the cloud in the radiative scheme of Carma were parameterized according to Sun and Shine (1994) and Savijärvi et al. (1997); Savijärvi and Räisänen (1998) using liquid and ice water content profiles provided by the JULES-CCATT-BRAMS cloud microphysics scheme (Freitas et al., 2017).





## 3 Results and discussion

### 3.1 Environmental characteristics in the Amazon basin scale

From the ECMWF ERA-interim reanalysis the evolution of the horizontal wind and air temperature near the surface, in the north region of Brazil, between July $6^{th}$ and $11^{th}$, 2014, at 12 UTC (Local Time = UTC - 4 h) (Fig.3) can be obtained. On the $6^{th}$ it is observed that the mean temperature was of the order of 24 $°C$ in three places of interest of this work, being: Porto Velho; Manaus and ATTO site (Fig. 3a). The dominant wind direction was from East in practically the entire Amazon region. The surface temperature and wind direction represent the standard normally found in this region (Fisch et al., 1998; Pöhlker

et al., 2019). However, on July $7^{th}$, the dominant wind direction becomes South-Southeast in the region of Porto Velho, as is evidenced by the presence of a mass of air with a lower temperature (around 18 $°C$) approaching this city (Fig. 3b).

In the course of the days, between July $8^{th}$ and $9^{th}$, the mass of cold air advanced even more towards Porto Velho, just as the dominant wind direction changed to South in all western regions of the state of Amazonas, as well as to the southern regions of Manaus and the ATTO site (Fig. 3c, d). On July $10^{th}$, the southerly winds arrive in the Manaus region and the

ATTO site, characterizing the arrival of Friagem in the area of interest of this work (Fig. 3e, f). For this period, the CPTEC technical bulletin reported the penetration of a polar air mass in the subtropical and tropical Brazilian region that advanced in the Southeast-Northwest of Brazil, giving origin to the cold waves of the South, as well as causing the Friagem phenomenon in the Amazon (http://tempo.cptec.inpe.br/boletimtecnico/pt).

Therefore, the arrival of the Friagem phenomenon in the Southwest and central regions of the Amazon is evidenced, that

produced abrupt drops in the values of temperature and change in the wind direction. Similar results were also found by other authors (Marengo et al., 1997; Fisch et al., 1998; de Oliveira et al., 2004).

The wind behavior throughout the Amazon basin before and during the Friagem event is represented in Fig. 4a and 4b, respectively. Interestingly, at the time the Friagem was present in the Manaus and ATTO site region, there was convergence of the easterly winds with the westerly flow associated to the Friagem (Fig. 4b). The easterly flow carries humidity from the

Atlantic coast to the central region of the Amazon, while the southerly flow, associated with the Friagem event, transports masses of dry and cold air from high latitudes to the Amazon region (Marengo et al., 1997).

Figure 5 shows the satellite images before and during the Friagem event in the central Amazon. Convection in the confluence between Amazonas and Tapajós rivers region was observed at dawn, on July 11 at 07 UTC (Fig 5a). This convection propagated in the West direction, arriving in the ATTO site region at 13 UTC (Fig 5c). Since this convective system is not associated to the

squall lines that form along the coast (Cohen et al., 1995; Alcântara et al., 2011; Melo et al., 2019) it is possible to state that this convection has its formation associated with the convergence of these two air masses with different properties (Fig 4). It is noteworthy that during the propagation of this convection on July $11^{th}$, it intensified and caused the highest rainfall (starting at 12:30 UTC) registered at the ATTO site during the month of July 2014, with a record rainfall of 21 $mm$.

The evolution of the horizontal wind and $O_3$-mixing ratio near the surface (both from ECMWF ERA-interim reanalysis),

during July $7^{th}$ and $11^{th}$, 2014, at 18 UTC can be seen in Fig. 6. On July $7^{th}$ onward the Friagem event carries air rich in $O_3$ northwards (Fig. 6a). This airmass reaches the state of the Amazonas on July $8^{th}$ (not show here). On July $11^{th}$ at 12 UTC (not



(a)

(b)

(c)

(d)

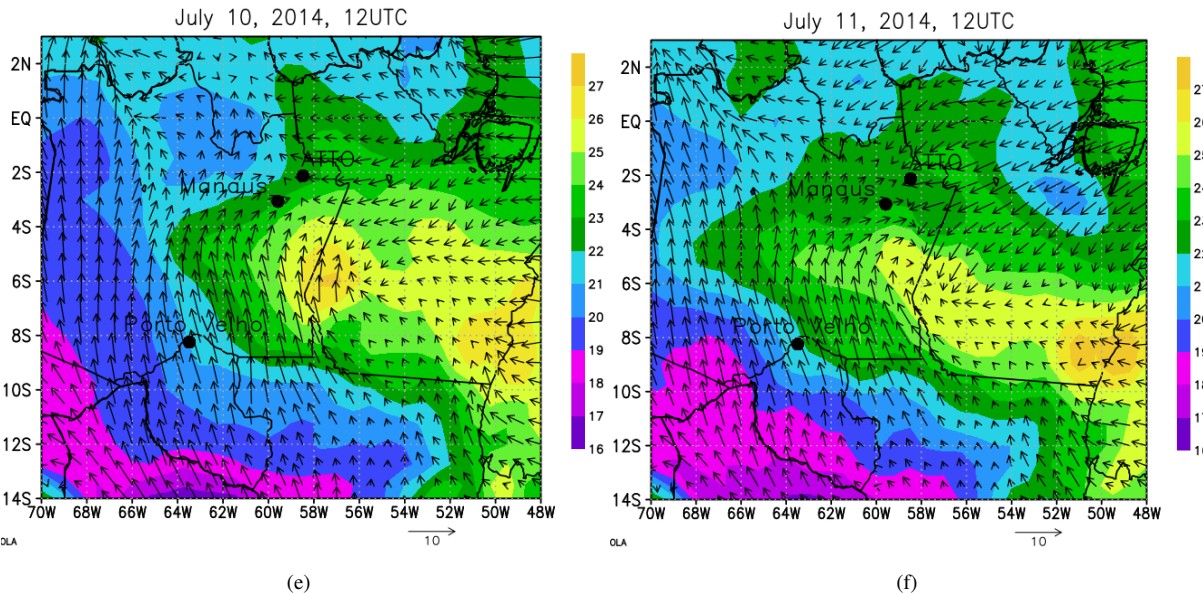

**Figure 3.** Distribution of air temperature ($^\circ C$, shaded) and wind ($ms^{-1}$, vector) at the surface, in the localities of Porto Velho, Manaus and ATTO site, at 12 UTC between days 6th and 11th of July 2014 obtained with the ERA-interim reanalysis.

shown here) the air mass influenced by the Frigaem has the shortest distance from the study region (ATTO-site). On July $11^{th}$ at 18 UTC the Friagem begins to dissipate (Fig. 6b). However, it should be noted that this mass of air rich in $O_3$ did not reach the Manaus region and the ATTO-site. It is believed that the presence of the cloud cover in central Amazonia on $11^{th}$, July (Fig. 5), formed by the convergence of air (Friagem and Eastern winds), has an inhibitory effect on $O_3$ formation (Betts et al., 2002). As $O_3$ deposition prevails, a net loss of ozone is expected during transport under conditions of limited photochemical production. The rain forest canopy is a strong sink for ozone (Jacob and Wofsy, 1990; Fan et al., 1990; Rummel et al., 2007). Therefore, the low $O_3$ mixing ratio in the Manaus region and the ATTO-site during the $11^{th}$ July (Fig. 6-f) would be associated with cloudiness and prolonged transport over forested regions.

Marengo et al. (1997) also showed that one of the effects of Friagem in central Amazonia is the induction of the cloudiness. They did not show the impact of Friagem on $O_3$ levels but showed that the values of income short radiation that reach the surface are greatly reduced during Friagem events. In this work we also find a strong reduction in income short radiation in central Amazonia that will be discussed in Fig. 10

## 3.2 Air temperature during the Friagem event

Figure 7 shows the air temperature values near the surface at Porto Velho (Fig 7a), Manaus region (Fig 7b) and above the forest canopy at the ATTO site (Fig 7c), between July $6^{th}$ and $11^{th}$, 2014 (black line) together with the air temperature hourly average for the month of July 2014 (orange line). At Porto Velho the difference between the maximum air temperature (before the arrival of the Friagem) and the minimum temperature (during the Friagem) decreased from $34\,^\circ C$ to $19\,^\circ C$ (decrease by

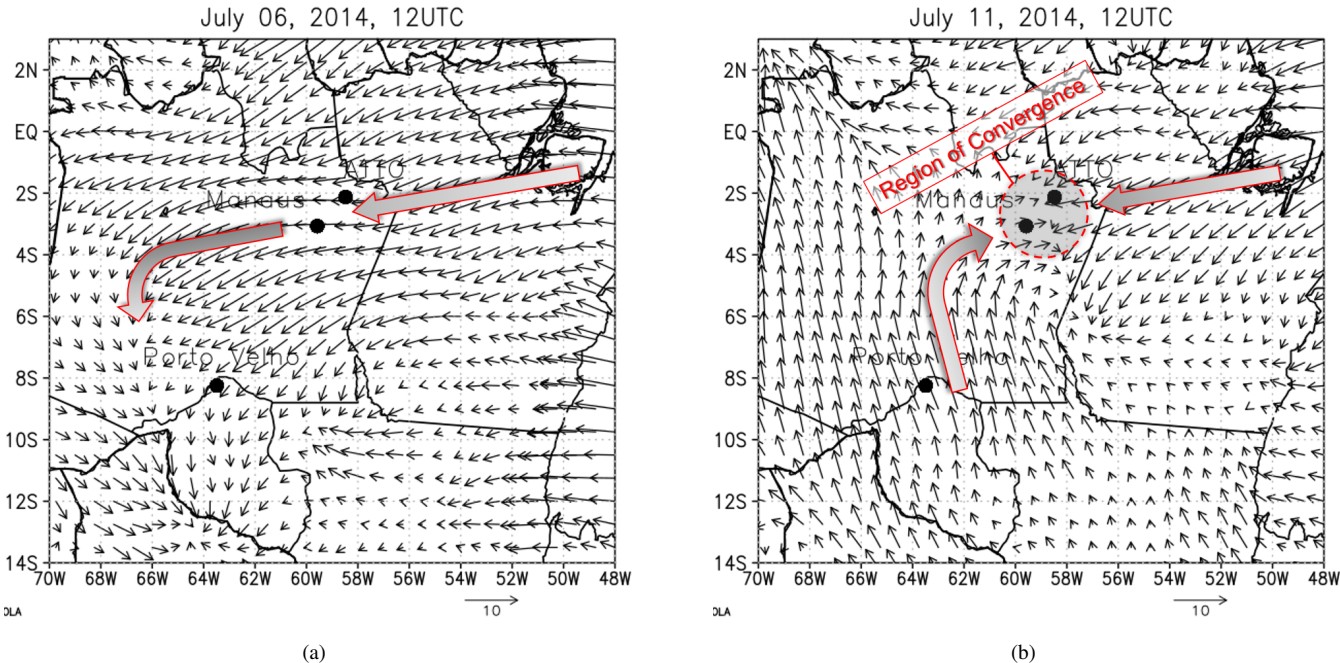

**Figure 4.** Surface wind behavior ($m\ s^{-1}$, vectors) on days (a) 6 and (b) 11 of July, 2014 at 12 UTC, highlighting Porto Velho, Manaus and ATTO site obtained with the ERA-interim reanalyses. Red arrows indicate the predominant wind flow and the dashed circle highlights the region of convergence of the winds in the Manaus and ATTO region.

15 °C) between July $7^{th}$ and $10^{th}$, whilst in Manaus region and at ATTO the decrease was in the order of 10 °C and 7 °C,
180 respectively, between the $9^{th}$ and $11^{th}$ of July. Therefore, the temperature fall in Manaus region and ATTO occurred around one day after the temperature fall observed in Porto Velho.

At Porto Velho, both the maximum and minimum values of air temperature were substantially reduced during the presence of the Friagem. However, at Manaus region and the ATTO site, the decrease was mainly observed in the maximum temperature values. Although the decrease was not so evident at the time of the diurnal minimum (at least on the $10^{th}$ and $11^{th}$) the whole
185 diurnal cycle was disturbed with (much lower) minima than the average at different times of the day.

Similar behavior was observed by Marengo et al. (1997) for the Southwest and Central Amazon regions during an episode of Friagem. Therefore, it is noted that due to the occurrence of the Friagem, the southernmost regions of the Amazon present more intense reductions in temperature values, compared to the regions located more in the center of the Amazon basin.

Additionally, the ATTO site is located in a forest region, $58\ km$ from the Balbina dam lake and Manaus region is under the
190 influence of intense urbanization (de Souza and Alvalá, 2014) and is located in the proximity of rivers. Thus, there is evidence that both the ATTO site and Manaus region may be under the influence of lake (Moura et al., 2004) and rivers breezes (dos Santos et al., 2014), respectively, which could offer them a greater thermal inertia.

**Figure 5.** Enhanced images of the GOES 13 satellite in the infrared channel on July $11^{th}$, 2014 at: (a) 07 UTC, (b) 11 UTC, (c) 13 UTC and (d) 16 UTC, which is openly accessible (http://satelite.cptec.inpe.br/acervo/goes.formulario.logic?i=br). Including the approximate locations of the ATTO site and Balbina lake (white circles) and the confluence region of the Amazon and Tapajós rivers (blue X).



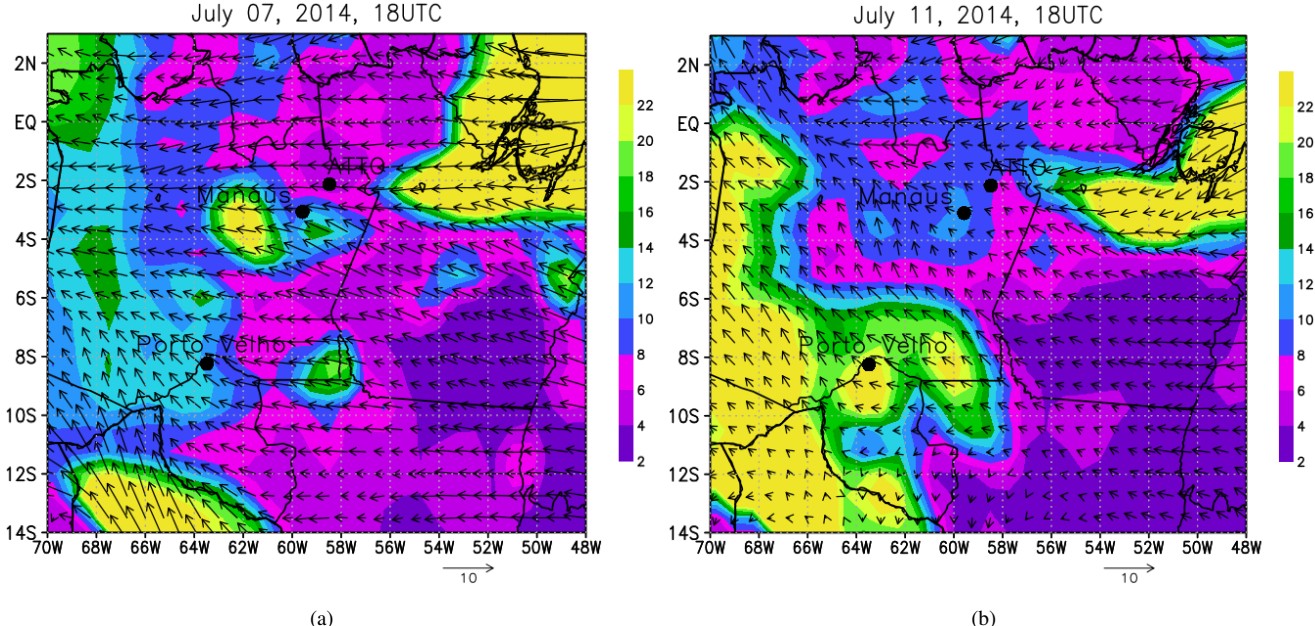

(a)                  (b)

**Figure 6.** Surface wind ($m\ s^{-1}$, vectors) and ozone (ppbv, contour) on days: (a) $7^{th}$ and $11^{th}$ of July, 2014 at 18 UTC, highlighting Porto Velho, Manaus and ATTO site obtained with the ERA-interim reanalysis.

### 3.3 ATTO site wind direction

In addition to the changes observed in the daily air temperature cycle at the ATTO site, changes were also observed in the local wind direction during the Friagem period (Fig 8). Before the arrival of this phenomenon, between July $6^{th}$ and $8^{th}$, it was observed that the direction of the horizontal wind was predominantly Southeast and Northeast. On the other hand, on July $9^{th}$ the wind direction was well distributed among the four cardinal points, and on July $10^{th}$ and $11^{th}$ the wind flow had higher frequencies of West, North and Southwest, when the Friagem arrived at ATTO site. The general wind directions before and after the Friagem are consistent with long term observations at ATTO (Andreae et al., 2015). The low frequency of observed wind directions from the westerly directions (based on 2.5 years of data) led to the conclusion that effects of local circulation (due to Uatumã River $\approx 12\ km$ and Balbina Lake $\approx 58\ km$) are not important or could not be observed (Andreae et al., 2015). At least not on regular basis.

Silva Dias et al. (2004) showed that during the arrival of a Friagem event in the Santarém-PA region (East of the Amazon), the atmospheric pressure at sea level increased, resulting in a pressure gradient force pointing in the opposite direction than the trade winds, which would be consistent with a slowdown of the easterly winds. In this way, these authors were able to observe with greater clarity the occurrence of river breeze circulations in this region. Following this hypothesis, the behavior of the wind at the ATTO site was analyzed every two hours, during the period in which the Friagem was active in this region Fig. 9.

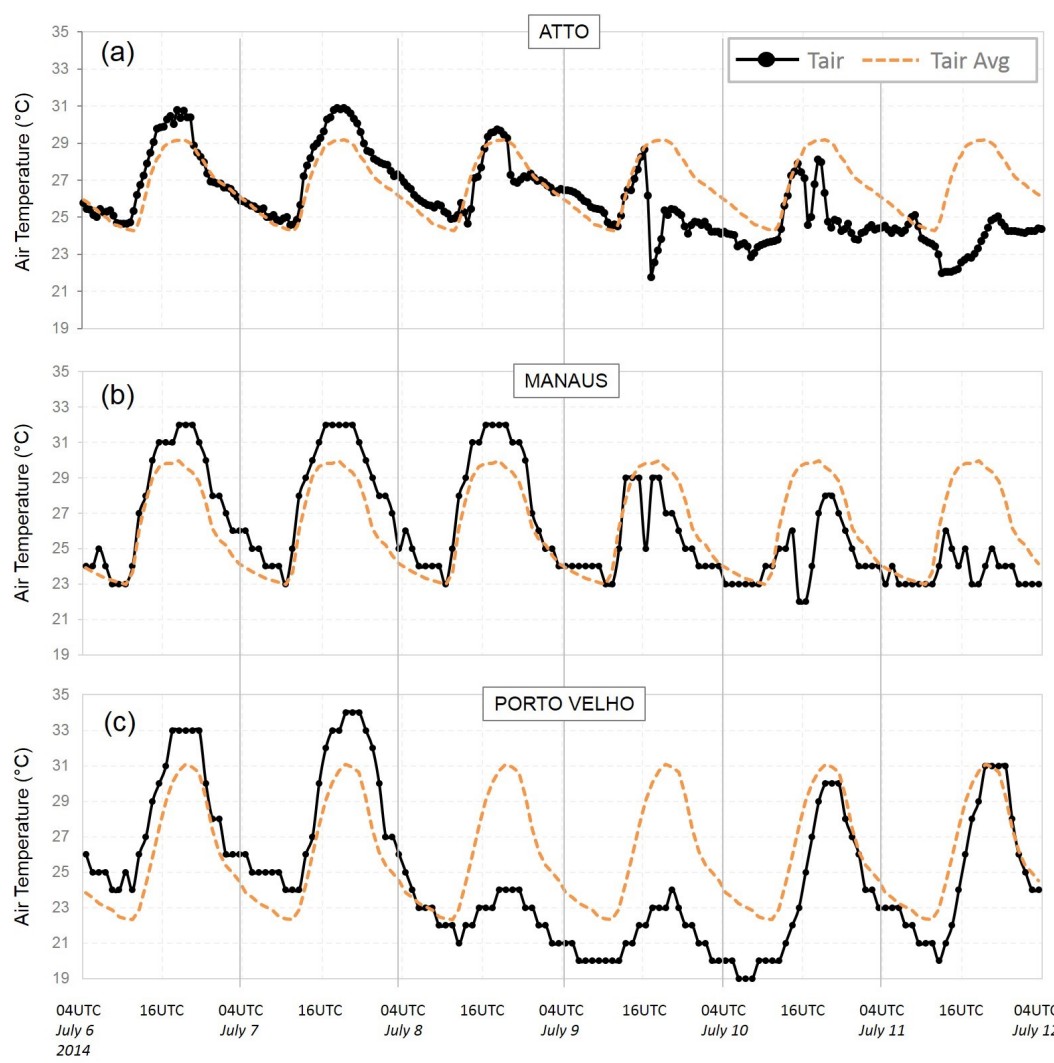

**Figure 7.** Daily cycle (black line) and monthly average (orange line) of the observational air temperature data from July $6^{th}$ to $11^{th}$, 2014, at: (a) ATTO site, (b) Manaus region and (c) Porto Velho.





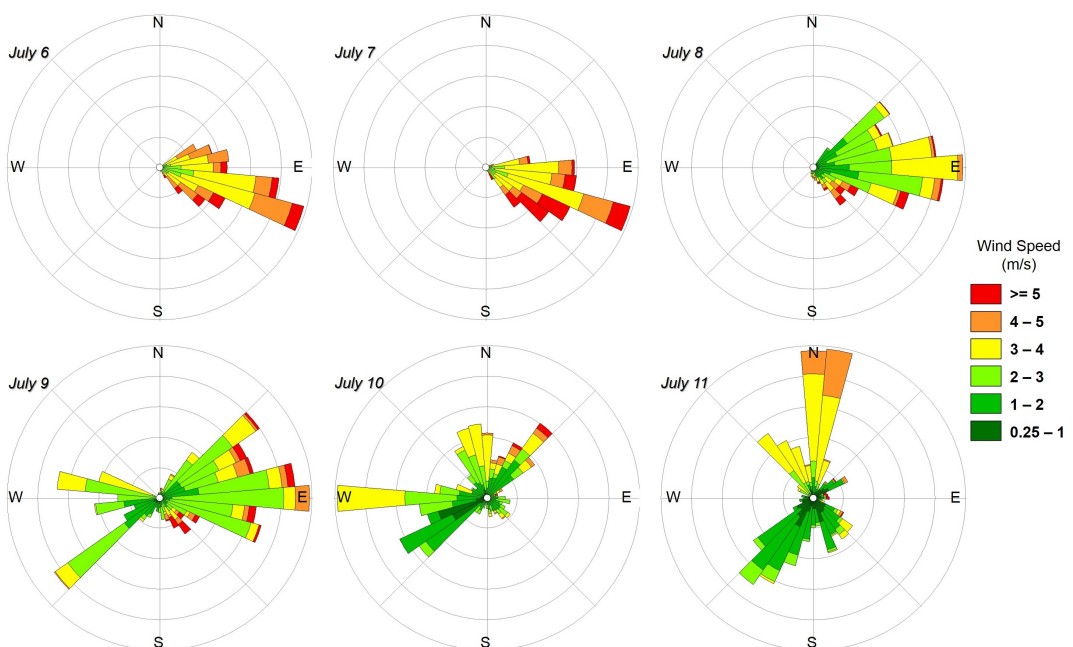

**Figure 8.** Horizontal wind speed and direction at $73\ m$ above ground measured at the ATTO site between July $6^{th}$ and $11^{th}$, 2014

On the $9^{th}$ of July it is observed that the direction of the wind was essentially from East until the end of the morning (14-16 UTC), when the wind changed to Southeast and Southwest directions until the late afternoon and early evening (22-00 UTC), which corresponds to the flow associated with the arrival of Friagem in this region. From 00 UTC of July $10^{th}$ to 14 UTC it is observed that the prevailing wind was from the West, indicating a deviation from the general flow, which would normally be from the East. In the early afternoon (16 UTC), the wind changed to the North direction until the early morning (12 UTC) of July $11^{th}$. This change in wind direction to the West and to the North observed during the morning of July $10^{th}$ and $11^{th}$, respectively, does not correspond to the expected direction during the occurrence of the forest breeze towards Lake Balbina. Therefore, it is believed that the flow related to the Friagem phenomenon overlapped with that of the breeze circulation observed by Moura et al. (2004), or that the forest-lake breeze circulation does not present the capacity to reach the micrometeorological tower of the ATTO site in $58\ km$ distance (In line with results from Andreae et al. (2015)). This aspect will be discussed in the next section where the results of the simulation with JULES-CCATT-BRAMS model will be analyzed.

### 3.4 Radiation, ozone and $CO_2$ during the Friagem event

Figure 10 shows the values of incident short wave radiation ($SW_{in}$), $O_3$ and $CO_2$ measured at the ATTO site, between July $6^{th}$ and $11^{th}$, 2014, respectively (black line). The $SW_{in}$ values decrease during the morning of July $11^{th}$ when Friagem arrives at the ATTO site (Fig 10a). Moreover, the maximum value ($\approx 450\ W\ m^{-2}$) of $SW_{in}$ occurred at approximately 19 UTC (15 LT), whereas the average monthly daily maximum $SW_{in}$ (orange line) usually occurs at 16 UTC ($\approx 800\ W\ m^{-2}$).



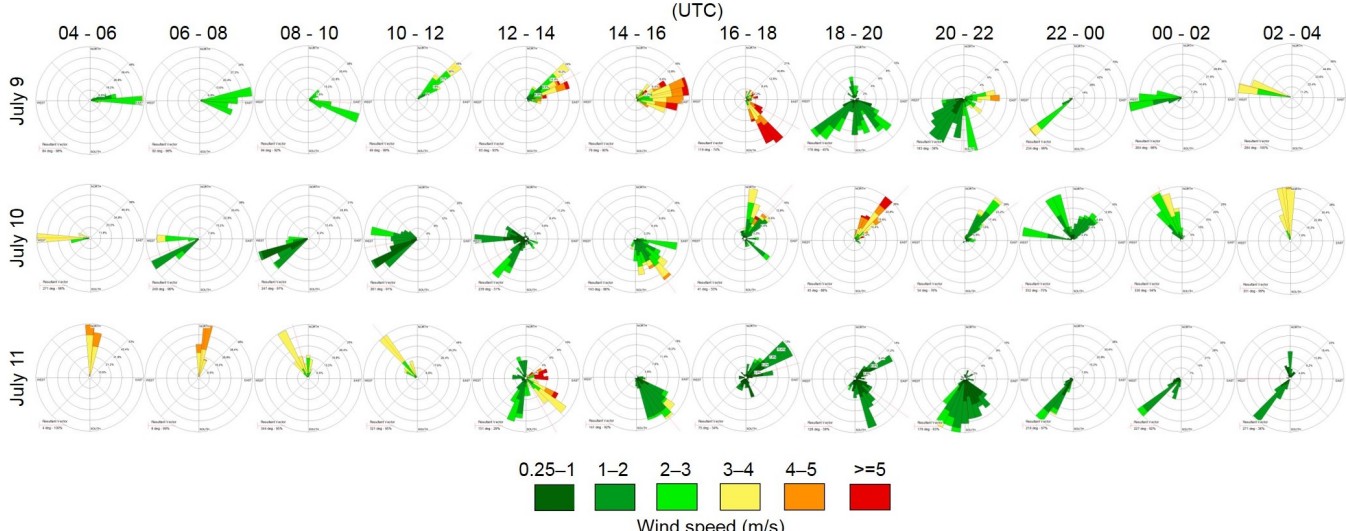

**Figure 9.** Wind speed and direction at the $73\ m$ above ground measured at the ATTO site, in 2 hour intervals, between July $9^{th}$ and $11^{th}$, 2014.

Before the arrival of Friagem at ATTO site region, between July $6^{th}$ and $8^{th}$, it is observed that the values of $O_3$ (black
line) were close to the monthly average (orange line), with minimum values occurring around 10 UTC (06 LT) and maximum
around 17 UTC (Fig 10b). This result is consistent with those observed in other studies conducted in the Amazon (Betts et al.,
2002; Gerken et al., 2016; Dias-Júnior et al., 2017; Melo et al., 2019). However, during the occurrence of Friagem, between
July $09^{th}$ and $11^{th}$, there was a sharp drop in $O_3$ mixing ratio at the times when the highest mixing ratio of this trace gas were
expected (17 UTC).

Figure 11 shows the $O_3$ mixing ratio data from 4 different stations around the city of Manaus. All stations show reduced $O_3$
values during the passage of the Friagem event (black dotted rectangle). Furthermore, stations affected directly by the pollution
of the city of Manaus (Iranduba - T2, Manacapuru - T3) show clear influence of increased $O_3$ formation compared to ATTO
and ZF2-T0z. These differences are much smaller during the Friagem event, probably due to reduced photochemistry (Fig.
10a) in this region.

The reduction of the incident short-wave radiation values observed on the $11^{th}$ (Fig 10a) was possibly associated to the
presence of convective systems in this region, as shown in Fig 5. It is known that cloudiness is a determinant meteorological
factor in the daily $O_3$ cycle (Gerken et al., 2016).

It is interesting to note that the rain event during July $11^{th}$ did not result in an increase of near surface $O_3$ as observed by
others authors (Betts et al., 2002; Gerken et al., 2016; Dias-Júnior et al., 2017). It is believed that the convective cloud formed
during the Friagem event was not as deep as the clouds investigated by Betts et al. (2002) and Gerken et al. (2016), which,
through their downdrafts, transport $O_3$ from the high troposphere to the surface.

The values of $CO_2$ mixing ratio between July $06^{th}$ and $11^{th}$ are shown in Fig 10c. It is observed that between July $06^{th}$ and $10^{th}$, $CO_2$ values for the daily cycle (black line) were very close to the monthly average values (orange line), with a maximum molar fraction around $420\,ppm$ approximately at 10 UTC and minimum of less than $390\,ppm$ (de Araújo et al., 2010). However, on July $11^{th}$ at 14 UTC, a significant increase of $CO_2$ ($\approx 470\,ppm$) was observed in relation to the monthly average. This increase may be related to the incident radiation attenuation due to increased cloudiness which reduces the efficiency of the forest in absorbing $CO_2$ gas via photosynthesis (Ruimy et al., 1995). Also limited vertical mixing as discussed below is a potential reason.

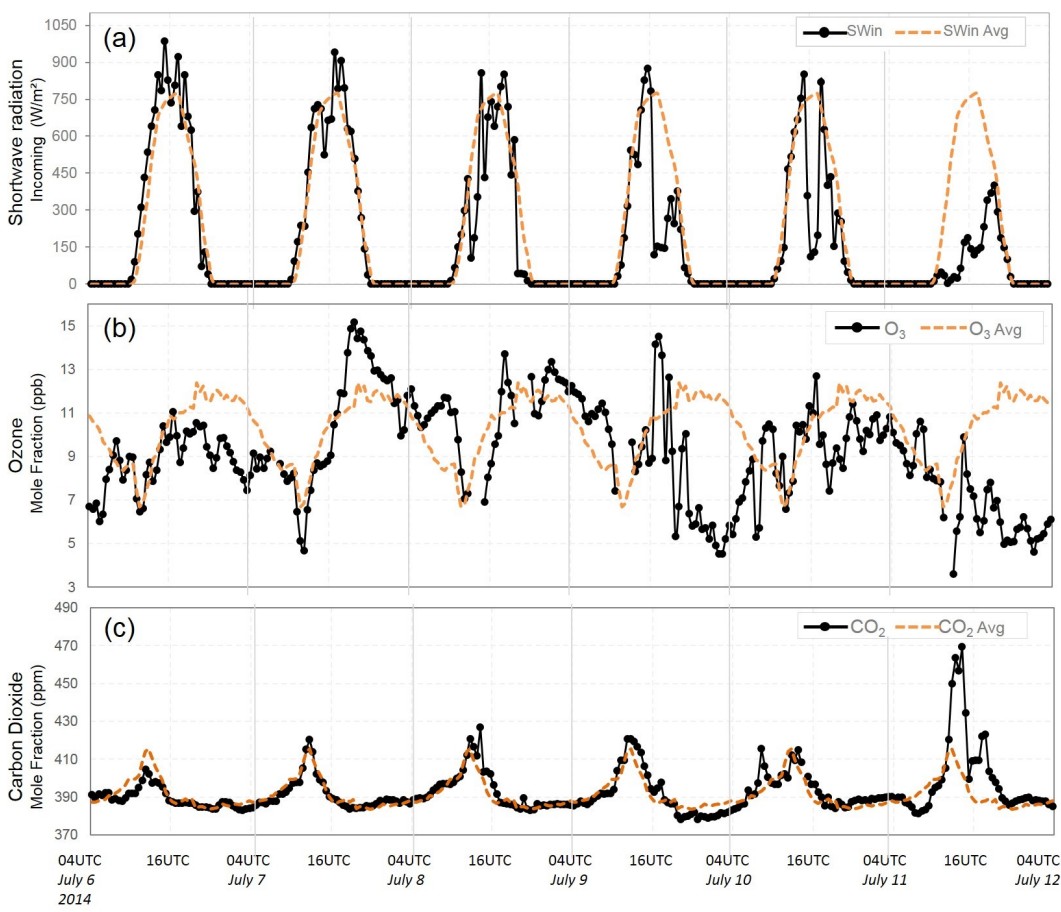

**Figure 10.** Daily behavior (black line) and monthly average (orange line) of incident short wave radiation ($SW_{in}$), Ozone ($O_3$) and Carbon Dioxide ($CO_2$) mixing ratio from July $6^{th}$ to $11^{th}$, 2014 at the ATTO site.





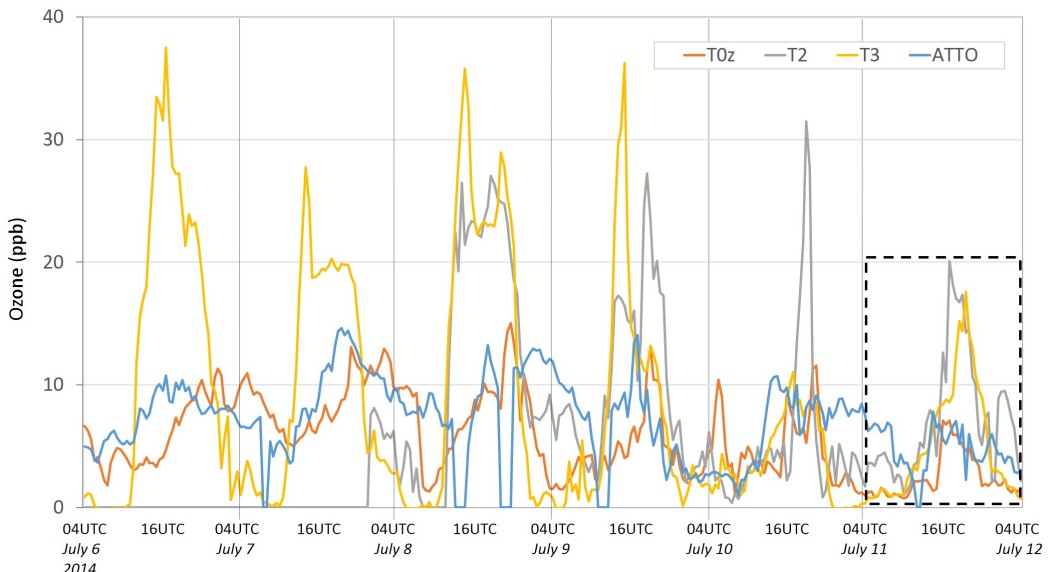

**Figure 11.** $O_3$-mixing ratio [ppbv] from 4 different stations around Manaus: ATTO-site (blue line), ZF2 forest - $T0z$ (red line), Iranduba - $T2$ (grey line), Manacapuru - $T3$ (yellow line). The black rectagle indicate the occurrence of the Friagem event. $T2$ are affected directly by the polluted air from the city of Manaus.

### 3.5 Simulation of local circulation and its effect at the ATTO site

In order to better understand the local circulation and its role on the measurements made at ATTO site region, this section presents the results of a numerical simulation made with JULES-CCATT-BRAMS coupled model. Figure 12a shows the vertical profile of the horizontal wind at a grid point near the ATTO site ($02°$ $S$ - $59°$ $W$) during model integration. At low levels (near $80$ $m$), the Easterly wind is observed until the first hours of July $10^{th}$. Then the wind has a predominant West-Northwest direction until the afternoon of July $11^{th}$ and afterwards the wind comes from the South. Therefore, it is observed that the sim-

ulation captured the horizontal wind behavior measured at a height of $73$ $m$ at the ATTO site, as shown in Fig 9. In addition, above $500$ $m$ the flow is essentially from the East during the whole period of integration of the model. Apparently, the Friagem changes only the flow within a small layer adjunct to the ground. Figure 12b shows the values of the boundary layer height (BLH) obtained form ERA5 at a grid point near the ATTO site ($02.10°$ $S$ - $59.06°$ $W$). It is possible to note that before the Friagem event the maximum BLH values were greater than 1000 m. However, during the Friagem event, the maximum BLH

value was around 600 m.

The large temperature drop Fig. 7a together with the information that the cold air of the Friagem was just in the lower $500$ $m$ (Fig. 12), points to the formations of a cold pool above the forest that prevents vertical mixing. As incoming solar radiation was low (Fig. 10a) the surface heating might not be sufficient to break the inversion or at least a very shallow boundary layer





was formed as evidenced by the ERA5 data (Fig.12b). This would explain high $CO_2$ (accumulation of soil emissions) and very

low $O_3$ (limited transport from aloft) at the same time at the ATTO site in addition to the reduced radiation (see section 3.3).

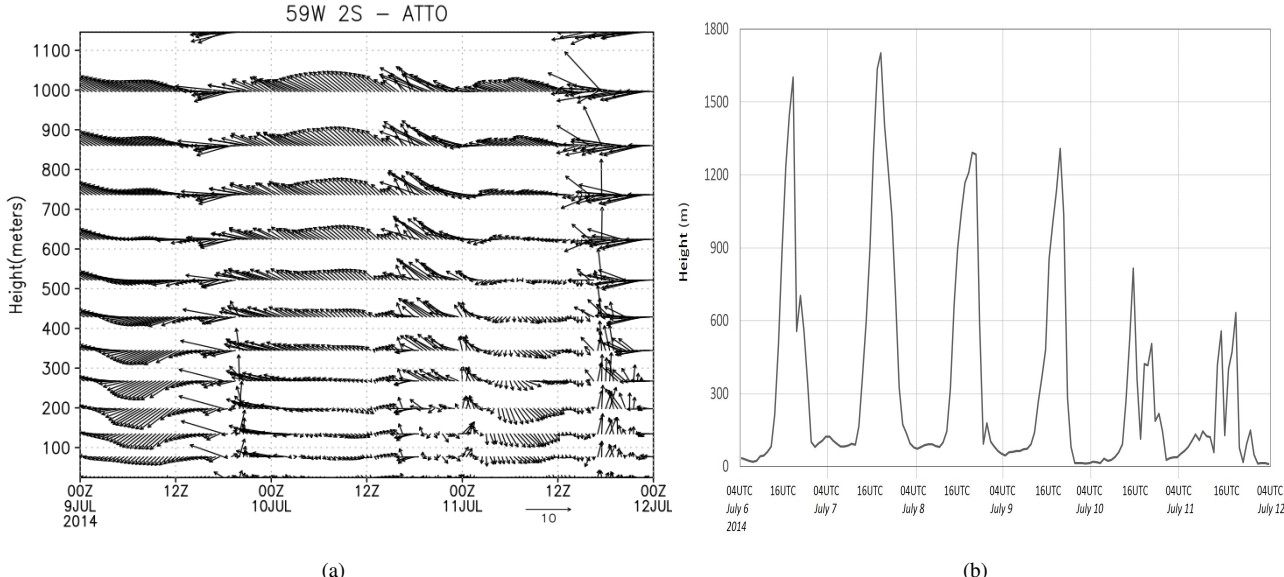

(a)                                                                      (b)

**Figure 12.** (a) Vertical profile of the horizontal wind ($m\ s^{-1}$) obtained by JULLES-CCATT-BRAMS simulation for the ATTO site from 00 UTC from the $9^{th}$ to 00 UTC of July $12^{th}$, 2014. (b) Boundary layer height ($m$) obtained form ERA5 for the ATTO site from 04 UTC - July $6^{th}$ to 04 UTC - July $12^{th}$, 2014.

Figure 13 shows the evolution of the temperature at $24.4\ m$ ($°C$, shaded) and horizontal wind ($m\ s^{-1}$, vector) at $134.5\ m$ on July $11^{th}$. Between 03 and 11 UTC, the air temperature is higher on Balbina Lake compared to that above the forest area. This temperature gradient induces the formation of a forest breeze towards the lake with the wind converging towards the center of the lake (Fig 13a-e). At 13 UTC the temperature gradient reverses its direction and induces the formation of the lake breeze

towards the forest that at 15 UTC is more clearly defined along the southeastern shores of Balbina Lake (Fig 13g).

Another interesting aspect is the entry of cooler air through the Northwest quadrant starting at 03 UTC, which is transported in Southeast direction. From 3 to 11 UTC a corridor of warmer air is established from Lake Balbina to the Southeast quadrant of the domain along the Uatumã River whose width is less than $1\ km$ and can not be captured by the horizontal resolution in this simulation. The gradual drop in temperature and predominance of Northwest winds shown in this simulation at the grid

points near the ATTO site agree with the observational data from this site (Fig. 7 and 8).

Although the Balbina lake breeze was established, it did not reach the ATTO site until 15 UTC (Fig 13g). In addition, precipitation in the simulation occurred in the following hours, similar to that observed in satellite images (Fig 5), which in turn disrupts the environment propitious to vigorous breezes that could reach the ATTO site. Although the Friagem phenomenon causes the weakening of the trade winds, which in turn would allow the establishment of more intense breezes as proposed

by Silva Dias et al. (2004), the cooler and drier air mass flow of Friagem in the central region of the Amazon was dominant





(a)

(b)

(c)

(d)





**Figure 13.** Evolution of modeled air temperature ($^{\circ}C$, shaded) at $24.4\ m$ and horizontal wind ($m\ s^{-1}$, vector) at $134.5\ m$, on July $11^{th}$, 2014 at: (a) 03 UTC, (b) 05 UTC, (c) 07 UTC, (d) 09 UTC, (e) 11 UTC, (f) 13 UTC, (g) 15 UTC and (h) 17 UTC. Balbina Lake (black contour) and ATTO site (black dot) are indicated





over the lake and forest breeze circulation. Possibly, the establishment of more vigorous river breeze circulations observed by Silva Dias et al. (2004) is possible due to the Friagem phenomenon not reaching that region and interfering with the signal of the breeze and causing intense rainfall.

Figure 14 shows the behavior of modeled water vapor, $O_3$, CO and $NO_2$ on July $11^{th}$, 2014, at the moment of incursion (a, c, f, g) and dissipation (b, d, f, h) of the Friagem in the study area. The mixing ratios of water vapor near the surface at 02 UTC (Fig. 14a) were lower in the regions where cooler air was observed entering this domain, indicating that the Friagem brought cold and dry air to the ATTO site and Balbina Lake.

$O_3$-mixing ratios are higher above the lake and its surroundings, for both times shown (Fig. 14c and d). The $O_3$-mixing ratio within the limits of the simulation domain are mostly below 11 ppbv, whereas above the lake these mixing ratios exceed 20 ppbv at certain points, especially at 02 UTC. The effect responsible for higher $O_3$-mixing ratio both during the day e night may be associated with the fact that deposition is very much reduced over the open water compared to the forest (Ganzeveld et al., 2009). It can also be seen that the Friagem extended in the direction of ATTO, but probably due to the onset of rain (Fig. 5) was not clearly detected at ATTO.

In regards of CO gas, it can be observed that its concentration on the center of the lake at 02 UTC (Fig. 14e) is higher than in the regions near the margins of the lake, however, calls attention at this time the transport of CO arriving with the South and Northeast winds, approaching the ATTO site. However, it is noted that the entire region of the simulation domain presents low CO mixing ratio at the time the Friagem is dissipated (Fig. 14f). Apparently, the Friagem event "expels" the polluted air mass in the South and Southeast of the ATTO site (around Manaus city), "cleaning" the atmosphere, or preventing this pollution from reaching ATTO site and Balbina lake.

$NO_2$ gas is an important precursor of $O_3$, and is mainly related to emissions from fires and vehicles. The emission of precursor gases in the formation of $O_3$ mixing ratio can increase of this trace gas to levels harmful to the forest, since the ozone can damage the stomatal functions of the leaves (Pacifico et al., 2015). In spite of this, it is observed that the higher $NO_2$ mixing ratio at 02 UTC (Fig. 14g) seem to have their origin in the region where higher $O_3$ mixing ratios are found and presented lower $NO_2$ during the time of dissipation of the Friagem (Fig. 14h).

## 4 Conclusion

In the period of July $9^{th}$ to $11^{th}$, 2014 a Friagem phenomenon reached the central region of the Amazon. Through the ECMWF ERA-interim reanalysis it was possible to verify that this phenomenon ventured the Amazon region from Southwest to Northeast, bringing a strong cold, dry, ozone-rich air mass in the West quadrant, which dominated the wind field in the central region of the Amazon.

Through the observational data it was possible to verify that the passage of the Friagem in central Amazon had its most significant effects on July $11^{th}$, in region of the Manaus city, such as: Balbina Lake; ATTO site and others sites (T2, T3 and T0z).




Water vapor July 11, 2014, 12UTC

(a)

Water vapor July 11, 2014, 12UTC

(b)

Ozone July 11, 2014, 02UTC

(c)

Ozone July 11, 2014, 12UTC

(d)



**Figure 14.** simulated horizontal wind at 134.5 m on July 11th, 2014 for: (a, b) water vapor mixture ratio ($g\ kg^{-1}$, shaded); (c, d) ozone mixing ratio (ppbv, shaded); (e, f) carbon monoxide mixing ratio (ppbv, shaded) and (g, h) nitrogen dioxide mixing ratio (ppbv, shaded) at 24.4 $m$, when the Friagem was arriving at the study area (a, c, e, g) and at the moment of its dissipation (b, d, f, g). Balbina Lake (black outline) and ATTO site (black dot) are indicated



From the observational data collected at the ATTO site, it was observed that the $11^{th}$ was marked by a sudden fall in air temperature, a weakening of the typical East flow and a predominance of South, West and North winds. In addition, on the $11^{th}$ the interaction between the Friagem air mass and the trade winds flow gave origin to convection bands, which in turn caused a significant reduction of the incident short wave radiation, besides a record rain of the month. With the BRAMS simulations we found that the cold air of the Friagem was just in the lower $500\ m$. These information leads us to the conclusion that there is a cold pool above the forest that prevents vertical mixing and consequently a increase in $CO_2$ mixing ratio and abrupt drop in $O_3$ mixing ratio is observed above the forest canopy.

Also, through the simulations of the JULES-CCATT-BRAMS it was possible to evaluate the main impacts that the Friagem phenomenon caused both in the thermodynamic characteristics and in the atmospheric chemistry of the central region of the Amazon. In general, the model reproduced satisfactorily the main changes that the phenomenon brought to the environment of interest. In addition, the breeze circulations between Lake Balbina and the forest were well represented in the simulations, however, it was not possible to verify the influence of this breeze in trace gas concentrations at the ATTO site.

With the observational results and the simulations, it can be concluded that the Friagem phenomenon can interfere deeply in the microclimatic conditions and the chemical composition of the atmosphere, in a region of dense forest, in the center of the Amazon.

*Data availability.* The ATTO data used in these study are stored in the ATTO databases at the Max Planck Institute for chemistry and the Instituto Nacional de Pesquisas da Amazônia. Data access can be requested from Stefan Wolff, who maintains the O3 mixing ratios dataset (stefan.wolff@mpic.de) and Alessandro Araújo who maintains the micrometeorology dataset (alessandro.araujo@gmail.com). The GoAmazon data used in these study can be requested from Luciana Rizzo (luvarizzo@gmail.com)

*Author contributions.* GFCN, JCPC and CQDJ designed the study and wrote the article with the assistance of MS and JHC. SW, RAFS and PA maintain the greenhouse gas measurement system at ATTO and provided the $CO_2$ and $O_3$ data. AA and MS operate and maintain the micrometeorology equipment at ATTO and provided the data which was fundamental for this study. LVR provided the O3 data from GoAmazon sites. PAFK and JCPC assisted with BRAMS simulations.

*Competing interests.* The authors declare that they have no conflict of interest

*Acknowledgements.* This work has been supported by the Max Planck Society (MPG). For the operation of the ATTO site, we acknowledge the support by the German Federal Ministry of Education and Research (BMBF contract nos. 01LB1001A and 01LK1602B) and the Brazilian Ministério da Ciência, Tecnologia e Inovação (MCTI/FINEP contract 01.11.01248.00) as well as the Amazon State University (UEA), FAPEAM, LBA/INPA, and SDS/CEUC/RDS-Uatumã. We also thank the European Centre for Medium-Range Weather Forecasts



(ECMWF) for the reanalysis data, and the Center for Weather Prediction and Climate Studies (Centro de Previso de Tempo e Estudos Climáticos (CPTEC/INPE)) for providing the JULES-CCATT-BRAMS model. Furthermore, the authors thank the Amazon Modeling Laboratory (Laboratório de Modelagem da Amazônia, LAMAZ) for the technical help given, and also the Brazilian Research Council (CNPq (Conselho Nacional de Desenvolvimento Científico e Tecnológico)) for the Masters scholarship. ERA-5 data are courtesy of Benedikt Steil



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
