# Peer review of "Friagem Event in Central Amazon and its Influence on Micrometeorological Variables and Atmospheric Chemistry"

_Atmospheric Chemistry and Physics, 2020_

## Referee Comment (RC1) · Anonymous Referee #1 · 26 Jul 2020

Friagem Event in Central Amazon and its Influence on Micrometeorological Variables and Atmospheric Chemistry

General comment: The manuscript presents an interesting discussion of how the entry of a cold front or cold can interfere with micrometeorological conditions and the rates of trace gas mixture in central Amazonia. The combination of surface measurements with the simulations of the coupled model JULES-CCATT-BRAMS made it possible to understand the cooling effects, as well as their development and implications. Certainly, the results related to the effects on Lake Balbina are important for understanding the effects of cold on the ecosystem as a whole. In general, the work has an important

scientific contribution, as it clearly and objectively shows the ecosystem's response to a cold event. With regard to the structure of the manuscript, it still needs adjustments in the text. Some structural modifications are needed to make it clearer to the reader around the methodological application used to achieve the proposed objectives. The only point to be reviewed more intensively is the choice of the study period and the implications of this in the discussions. As the methodology of the work itself shows, this manuscript brings as results the case study of a particular event that occurred from July 6 to 11, 2014, however, no discussion about the meteorological characteristics of this year was held, it was also not clear whether any cold front arrival in the region will cause the same effects. The authors cite other studies on coldness in the Amazon, which are in agreement with their results, but do not make clear when these analyses were performed. As much of the results are derived from simulations it would be interesting to discuss the possible annual variations or at least discuss whether such variations may exist or not, as well as answer whether the effects on atmospheric chemistry will always be these, or if by different conditions, such as a year with high burn rates, these results may diverge, that is, my suggestion is a small restructuring of the results to include these discussions.

Specific comments: About the abstract: Review the first sentence of the abstract, because it practically already brings, in a more generic way, the main conclusion of the work, that is, the authors begin the work stating that the cold event influences the variables and atmospheric chemistry. I suggest changing the sentence and leaving to make this statement at the end of the abstract along with the main conclusions of the work.

About the introduction: In paragraph 30, the authors evidence the influence of breezes on $CO_2$ and $O_3$ mixing rates, however, they mention a region of North America, Canada, and this is out of context in the manuscript because all other information collected in the introduction directly mentions works developed in the Amazon. If the authors want to talk more about these events around the world, they should include

supplementary discussions on the effects of lake breezes. The last sentence of paragraph 50 is a text that describes how the objectives will be achieved, that is, a text of methodology, I suggest removing or restructuring this text since this information will appear in the methodology.

About the methodology: In paragraph 70 the authors say that this is a case study, it would be interesting at this moment to talk about the specific implications of this analyzed period. When talking about the O3 measurements in the analyzed sites, it is observed that these measurements were performed at different heights, ATTO at 79m, T3 at 3.5m, T2 at 12m and T0z at 39m. Can these different heights interfere with the measurements? The authors can make a brief discussion about this.

On the results: the results are presented in a very clear and objective way, the only observation is made in relation to the period of analysis. As described in the methodology of the work, this manuscript brings as results the case study of a particular event that occurred from July 6 to 11, 2014, however, no discussion about the meteorological characteristics of this year was held, it was also not clear whether any cold front arrival in the region will cause the same effects. The authors cite other studies on coldness on Amazon, which are in agreement with their results, but do not make clear when these analyses were performed. As much of the results are derived from simulations it would be interesting to discuss the possible annual variations or at least discuss whether such variations may exist or not, as well as answer whether the effects on atmospheric chemistry will always be these, or if by different conditions, such as a year with high burn rates, these results may be different, that is, I suggest a small restructuring of the results so that these discussions are included.

About the figures presented in the results: In general, give more detailed information of the figures in the subtitles. The figures along with their subtitles have to be high-explanatory. Another detail that the authors have to review are the titles of the axes of the figures, as well as the title in the "colobar" when necessary.

On the conclusion: In paragraph 320 the authors state that in general, the model satisfactorily reproduced the main changes caused by the cold phenomenon. Did the authors intend to evaluate the application of the model? Was that a goal, too? Just one observation in the last sentence of the conclusion: it is practically the same initial sentence in the abstract, so is necessary to restructure this fragment in the abstract.
* * *

---

## Referee Comment (RC2) · Anonymous Referee #2 · 10 Aug 2020

General comments The manuscript studies a Friagem event during July 9 – 11, 2014 in the central Amazon region and its influences on the micrometeorology variables, local circulation, as well as the trace gas concentrations. The investigation of a cold front in the central Amazon is a relevant subject for research in current days. Using the reanalysis and the satellite data, the manuscript demonstrates the propagation of the cold front and the convection on Jul 11 2014. The second main component of the paper is to understand the event mechanistically and its influences with the local circulation by simulating the cold front. The third component is to explore the influences of this front on the temperature and the trace gas concentrations. I trust most of the results regarding the meteorological part such as the occurrence of the cold front and its link

to the convection on Jul 11. I feel the weaknesses of the manuscript is the depth of discussion and the interpretation of the chemistry part.

The cold front has a lifetime of 3∼ 5 days as presented in the manuscript, while O3 has a much shorter lifetime. It is tricky to quantitively define the influences of the cold front on O3 directly due to their different timescales. Specifically, the authors suggest that the cold pool arrives at ATTO on July 9-11. However, the O3 mixing ratios are affected on the 9th and 11th by convective systems, not on the 10th. To me the O3 concentrations are closely related to the convective systems not the cold pool necessarily. In addition, the dry deposition and vertical mixing are heavily speculated to play a part in the O3 concentrations without actually being estimated.

The general features of the cold front are clearly described in the manuscript such as the temperature drops and the trade wind is weakened, which accounts for the majority of the manuscript. However, the understanding and discussion of its mechanism is lacking. For example, it is not clear how the cold front induces the convection on July 11 that affects the O3, and thus it's still unclear to what extent Friagem affects O3 in general without knowing its influences on inducing convections. In addition, the cold pool and the subsequent weakened vertical mixing are not well demonstrated because of the lack of vertical profiles of meteorological variables. I believe these can be fixed by further exploring the model results.

Major comments 1. Line 145: Figure 3 suggest that the changes in temperature are not that significant for Manaus and ATTO, somewhere within 2 degrees. 2. Figure 4: Is the same data in Fig. 4 as in Fig. 3a and 3f? I wouldn't show the same data twice. 3. Line 160 "carries air rich in O3": What is (are) the source(s) of the O3? 4. Line 166-167: The chemical reactions with terpenes emitted by the forest might be important for O3 loss too. The estimate of the lifetime of the O3, which is a function of dry deposition and chemical reactions is needed for this argument "As O3 deposition prevails, a net loss of ozone is expected during transport under conditions of limited photochemical production". 5. Line 178-179: How the maximum air temperature is defined here?

[Figure]

Seems like it is part of the diurnal cycles, which to me is not an appropriate metric for evaluating the intensity of the Friagem. 6. Line 213-215: not clear. Clarify. 7. Line 228: Any explanations for the decreases in O3? 8. Line 238-241: "did not result in an increase of near surface O3". I don't necessarily agree with this. I think there is an increase in O3 from roughly 6 ppbv to 10 ppbv. To validate if this increase is due to the convection, you can calculate the virtual potential temperature as in Gerken et al. (2016). 9. Line 247 and 262: The vertical mixing can be evaluated by the vertical profiles of the virtual potential temperature. 10. Figure 13: Why the temperature at 24.4 m is used? It is within the canopy if I understand correctly, which I think would be very different (presumably lower) from above-canopy temperature. 11. How well the surface layer is represented by the JULES-CCATT-BRAMS model in general? How about in this study? Any comparisons between the modelled and the observations to evaluate the fidelity of the model for surface layer? 12. Line 318-319: The suppressed vertical mixing might play a part in the decreased O3 mixing ratios, but it's not the only or main reason here.

Minor comments 1. Line 66: I'd cite more relevant studies regarding O3 at the T3 site. 2. Line 67: I'd point out the minimal anthropogenic influences at the ZF2 site to contrast the other sites. 3. Table 1: What is the canopy height at ATTO site? 4. Figure 7: I'd present the data in the order of Porto Velho, Manaus, and ATTO. 5. Line 207: There are some editorial/technical issues to be fixed. For example, the parentheses are missing for "Fig. 9".

---

## Author Comment (AC1) · 21 Oct 2020

Question

General comments: The manuscript presents an interesting discussion of how the entry of a cold front or cold can interfere with micrometeorological conditions and the rates of trace gas mixture in central Amazonia. The combination of surface measurements with the simulations of the coupled model JULES-CCATT-BRAMS made it possible to understand the cooling effects, as well as their development and implications. Certainly, the results related to the effects on Lake Balbina are important for understanding the effects of cold on the ecosystem as a whole. In general, the work has an importante

scientific contribution, as it clearly and objectively shows the ecosystem's response to a cold event. With regard to the structure of the manuscript, it still needs adjustments in the text. Some structural modifications are needed to make it clearer to the reader around the methodological application used to achieve the proposed objectives. (1) The only point to be reviewed more intensively is the choice of the study period and the implications of this in the discussions. As the methodology of the work itself shows, this manuscript brings as results the case study of a particular event that occurred from July 6 to 11, 2014, however, no discussion about the meteorological characteristics of this year was held, it was also not clear whether any cold front arrival in the region will cause the same effects. The authors cite other studies on coldness in the Amazon, which are in agreement with their results, but do not make clear when these analyses were performed. (2) As much of the results are derived from simulations it would be interesting to discuss the possible annual variations or at least discuss whether such variations may exist or not, as well as answer whether the effects on atmospheric chemistry will always be these, or if by different conditions, such as a year with high burn rates, these results may diverge, that is, my suggestion is a small restructuring of the results to include these discussions.

Answer

We appreciate the reviewer's comments. We will respond in parts: (1): The reasons for choosing the case study shown in the manuscript (July 6 to 11, 2014), were as follows: i) July is one of the months with the largest number of cold fronts that arrive in the South-Southeastern region of Brazil (Prince and Evans, 2018). Consequently, July is also the month where a greater number of Friagem phenomena are observed in the Amazon region (Prince and Evans, 2018). ii) Throughout 2014, intensive activities of the GoAmazon project took place (Martin et al., 2016), that is, measurements of gases and the thermodynamics of the atmosphere were carried out in several sites investigated in this work (T2, T3 and T0z), and therefore this was the motivation for choosing the year 2014 for our case study. iii) The period between 06 and 11 July was

chosen, as it was observed that a Friagem event reached the city of Manaus and its surroundings in those days. It should be noted that for a Friagem event to occur, it is necessary that a mass of cold air (cold front), coming from the South reaches the North region of Brazil. Friagem events do not always have the "capacity" to reach the city of Manaus. For example, on July 25-31 2014 there was also a Friagem event in the Southwest of the Amazon, but this event was not observed in the city of Manaus. About the meteorological characteristics of this year, according to the CLIMANALISE Bulletin (http://climanalise.cptec.inpe.br/~rclimanl/boletim/pdf/pdf14/jul14.pdf), in July 2014, precipitation in northern Brazil showed positive and negative deviations from the climatological average (Figure 1a). In addition, the deviation from the maximum temperature in relation to its climatology shows a drop in the maximum temperature from the state of São Paulo to the Southwest of the Amazon, indicating the advance of frontal systems in this region (Figure 1b). Regarding global scale phenomena, the South Oscillation Index showed that this month remained close to neutral, that is, without the occurrence of the El Nino and La Nina phenomena. The main characteristics of the Friagem observed in this work seem very similar to those observed by Marengo et al. (1997) and Silva-Dias et al. (2004), both cited in the manuscript. Marengo et al. (1997) investigated the two strongest Friagem events that occurred during the year 1994, being: June 26th and July 10th. For both events they observed that the main consequence of the Friagem in the City of Manaus was greater cloud cover and consequently less solar radiation reaching the surface, which is the main cause of the fall in air temperature. In addition, they noted that Friagens produced a shallower boundary layer. That is, the results by Marengo et al. (1997) corroborate part of our results - Friagem increases the cloud cover (Fig. 4), reduces the air temperature (Fig. 6) and produces a shallower boundary layer (Fig. 11a). The work by Silva-Dias et al. (2004) showed that during the period from 24 to 31 July 2001, the arrival of a cold air mass in the western region of the Amazon increased atmospheric pressure to sea level in this region, resulting in a pressure gradient force pointing in the opposite direction of the trade winds, which is consistent with a deceleration of the trade winds and the

consequent formation of more intense breeze circulations in the Santarém region. The main consequences of this Friagem in the city of Manaus were: drop in air temperature around 5 °C, reduction in wind speed, confluence of a cold and dry air mass coming from the South region with a hot and humid air mass coming eastern Amazon. We emphasize that part of our results are corroborated by Silva-Dias et al. (2004), which are: (1) confluence of trade winds with westerly winds in central Amazonia (Fig. 3). We show that it was this confluence that was mainly responsible for the formation of clouds and the consequent reduction of solar radiation that reached surfaces, reducing the air temperature and the O3 concentration. (2): We agree with the reviewer that new simulations that show the impact of possible annual variations, such as the increase/decrease in precipitation and air humidity and decrease/increase in temperature, during atypical years, such as La Niña/El niño, among others, can influence the number of occurrences and the strength of Friagem events and, consequently, the chemistry and thermodynamics of the atmosphere near the surface. In addition, the performance of simulations with different burn rates conditions and consequently with different amounts of cloud condensation nuclei can influence the formation of clouds and the role of cooling above the central Amazon. However, the objective of this work is not to make comparisons between different annual conditions, but to demean a case study. The reviewer's suggestions are valuable and will be the subject of future research by this group. In addition, we will add these suggestions to the conclusions of the manuscript (suggestions for future work).

Question

Specific comments: About the abstract: Review the first sentence of the abstract, because it practically already brings, in a more generic way, the main conclusion of the work, that is, the authors begin the work stating that the cold event influences the variables and atmospheric chemistry. I suggest changing the sentence and leaving to make this statement at the end of the abstract along with the main conclusions of the work.

Answer

We decided to move this sentence from the abstract to the conclusions section.

Question

About the introduction: In paragraph 30, the authors evidence the influence of breezes on $CO_2$ and $O_3$ mixing rates, however, they mention a region of North America, Canada, and this is out of context in the manuscript because all other information collected in the introduction directly mentions works developed in the Amazon. If the authors want to talk more about these events around the world, they should include supplementary discussions on the effects of lake breezes. The last sentence of paragraph 50 is a text that describes how the objectives will be achieved, that is, a text of methodology, I suggest removing or restructuring this text since this information will appear in the methodology.

Answer

We agree with the reviewer: We rewrite the paragraph 30 and we remove the last sentence of paragraph 50 that described how the objectives will be achieved.

Question

About the methodology: In paragraph 70 the authors say that this is a case study, it would be interesting at this moment to talk about the specific implications of this analyzed period.

Answer

We introduced a new paragraph to better explain the motivation for choosing July 2014 as case study and we made a brief comment about the specific implications of this analyzed period (L68-75).

Question

When talking about the O3 measurements in the analyzed sites, it is observed that these measurements were performed at different heights, ATTO at 79m, T3 at 3.5m, T2 at 12m and T0z at 39m. Can these different heights interfere with the measurements? The authors can make a brief discussion about this.

Answer

Yes, different measurement heights may affect the observed O3 concentrations in some cases, due to the process of dry deposition onto available surfaces and stomatal uptake by vegetation. In the case of T2 and T3 sites, which are not forest sites, the measurement height may not have a significant influence on O3 concentrations during the day in a well mixed boundary layer, provided that the inlets were set apart from surfaces like walls, roofs and trees. At forest sites, previous studies have shown a significant O3 vertical gradient inside the canopy, especially in its lowest half part (e.g., Rummel et al., 2007; Freire et al., 2017). However, the reported O3 measurements at T0z and ATTO were taken above the canopy, where vertical gradients are expected to be close to zero if the boundary layer is well mixed. Based on previous studies, we estimate that the 40 m difference in the measurement height of ATTO and T0z may result in a 15% difference on O3 concentrations, with smaller concentrations at T0z due to the proximity of the canopy top. Nevertheless, this difference does not affect the main aspect discussed in Figure 11, which clearly shows a decrease in diurnal O3 concentrations at all sites in 2014 July 11th as a result of the influence of a cold front. We put part of this comment in the main text of the manuscript (L95-101).

Question

On the results: the results are presented in a very clear and objective way, the only observation is made in relation to the period of analysis. As described in the methodology of the work, this manuscript brings as results the case study of a particular event that occurred from July 6 to 11, 2014, however, no discussion about the meteorological characteristics of this year was held, it was also not clear whether any cold front arrival

in the region will cause the same effects. The authors cite other studies on coldness on Amazon, which are in agreement with their results, but do not make clear when these analyses were performed.

Answer

We inserted new paragraphs in the manuscript that make the meteorological characteristics of this year (L68-75) and in our citations about other studies on coldness on Amazon we make more clear when these analyzes were performed (L181-184; L214-218)

Question

As much of the results are derived from simulations it would be interesting to discuss the possible annual variations or at least discuss whether such variations may exist or not, as well as answer whether the effects on atmospheric chemistry will always be these, or if by different conditions, such as a year with high burn rates, these results may be different, that is, I suggest a small restructuring of the results so that these discussions are included.

Answer

We agree with the reviewer that new simulations that show the impact of possible annual variations, such as the increase/decrease in precipitation and air humidity and decrease/increase in temperature, during atypical years, such as La Niña/El niño, among others, can influence the number of occurrences and the strength of Friagem events and, consequently, the chemistry and thermodynamics of the atmosphere near the surface. In addition, the performance of simulations with different burn rates conditions and consequently with different amounts of cloud condensation nuclei can influence the formation of clouds and the role of cooling above the central Amazon. However, the objective of this work is not to make comparisons between different annual conditions, but to demean a case study. The reviewer's suggestions are valuable and will be

the subject of future research by this group. In addition, we will add these suggestions to the conclusions of the manuscript (suggestions for future work).

Question

About the figures presented in the results: In general, give more detailed information of the figures in the subtitles. The figures along with their subtitles have to be highexplanatory. Another detail that the authors have to review are the titles of the axes of the figures, as well as the title in the "colobar" when necessary.

Answer

Thank you. We reviewed the figure captions and made some minor changes (in blue). In all the figures where there is "colobar" we indicate that they represents the shaded area. The axes that do not have a title are those that indicate the North/South and East/West coordinates.

Question

On the conclusion: In paragraph 320 the authors state that in general, the model satisfactorily reproduced the main changes caused by the cold phenomenon. Did the authors intend to evaluate the application of the model? Was that a goal, too? Just one observation in the last sentence of the conclusion: it is practically the same initial sentence in the abstract, so is necessary to restructure this fragment in the abstract.

Answer

We would like to thank the reviewer for his comments. We decided to remove the sentence "In general, the model reproduced satisfactorily the main changes that the phenomenon brought to the environment of interest" from the conclusion and the sentence "that is, the Friagem event has the ability to significantly change the microclimate and atmospheric chemistry close to the surface in the Amazon central region" of the abstract.

References

Freire, L. S., Gerken, T., Ruiz‐Plancarte, J., Wei, D., Fuentes, J. D., Katul, G. G., Dias, N. L., Acevedo, O. C., and Chamecki, M. Turbulent mixing and removal of ozone within an Amazon rainforest canopy, J. Geophys. Res. Atmos., 122, 2791– 2811, doi:10.1002/2016JD026009, 2017.

Marengo, J. A., Nobre, C. A., and Culf, A. D.: Climatic impacts of "friagens" in forested and deforested areas of the Amazon basin, J. Appl. Meteorol., 36, 1553–1566, https://doi.org/10.1175/1520-0450(1997)036<1553:CIOFIF>2.0.CO;2, 1997.

Martin, S. T., Artaxo, P., Machado, L. A. T., Manzi, A. O., Souza, R. A. F., Schumacher, C., Wang, J., Andreae, M. O., Barbosa, H. M. J., Fan, J., Fisch, G., Goldstein, A. H., Guenther, A., Jimenez, J. L., Pschl, U., Silva Dias, M., Smith, J. N., and Wendisch, M.: Introduction: Obser-vations and Modeling of the Green Ocean Amazon (GoA-mazon2014/5), Atmos. Chem. Phys., 16, 4785–4797, https://doi.org/10.5194/acp-16-4785-2016, 2016.

Prince, K. C. and Evans, C.: A Climatology of Extreme South American Andean Cold Surges, J. Appl. Meteorol. and Climatol., 57, 2297–2315, https://doi.org/10.1175/JAMC-D-18-0146.1, 2018.

Rummel, U., Ammann, C., Kirkman, G., Moura, M., Foken, T., Andreae, M., and Meixner, F.: Seasonal variation of ozone deposition to a tropical rain forest in southwest Amazonia, Atmos. Chem. Phys., 7, 5415–5435, https://doi.org/10.5194/acp-7-5415-2007, 2007.

Silva Dias, M., Dias, P. S., Longo, M., Fitzjarrald, D. R., and Denning, A. S.: River breeze circulation in eastern Amazonia: observations and modelling results, Theor. Appl. Climatol., 78, 111–121, https://doi.org/10.1007/s00704-004-0047-6, 2004.

Please also note the supplement to this comment:

https://acp.copernicus.org/preprints/acp-2020-564/acp-2020-564-AC1-supplement.pdf

[Figure]

**(a) Deviation of precipitation in July 2014**

**(b) Maximum temperature deviation (ºC) in July 2014**

Figure 1. Behavior (a) deviation of accumulated precipitation in relation to climatological-mean (1961-1990) and (b) deviation from maximum temperature in relation to climatological-mean (1961-1990) for July 2014.

Source: Monitoring and Climate Analysis Bulletin (CLIMANASE). V. 29, No.07, July 2014. ISSN 0103-0019 CDU-555.5

**Fig. 1.**

---

## Author Comment (AC2) · 21 Oct 2020

General comments: The manuscript studies a Friagem event during July 9 - 11, 2014 in the central Amazon region and its influences on the micrometeorology variables, local circulation, as well as the trace gas concentrations. The investigation of a cold front in the central Amazon is a relevant subject for research in current days. Using the reanalysis and the satellite data, the manuscript demonstrates the propagation of the cold front and the convection on Jul 11, 2014. The second main component of the paper is to understand the event mechanistically and its influences with the local circulation by simulating the cold front. The third component is to explore the influences

of this front on the temperature and the trace gas concentrations. I trust most of the results regarding the meteorological part such as the occurrence of the cold front and its link to the convection on Jul 11. I feel the weaknesses of the manuscript is the depth of discussion and the interpretation of the chemistry part.

(1) The cold front has a lifetime of  $3 \sim 5$  days as presented in the manuscript, while O3 has a much shorter lifetime. It is tricky to quantitively define the influences of the cold front on O3 directly due to their different timescales. Specifically, the authors suggest that the cold pool arrives at ATTO on July 9-11. However, (2) the O3 mixing ratios are affected on the 9th and 11th by convective systems, not on the 10th. (3) To me the O3 concentrations are closely related to the convective systems not the cold pool necessarily. (4) In addition, the dry deposition and vertical mixing are heavily speculated to play a part in the O3 concentrations without actually being estimated.

**Answer**

We thank the reviewer for pointing this out. We will take the opportunity to better explain the results. We will answer in 4 parts: (1) This is consistent with our argumentation!! From model results, we see higher O3 concentrations associated with the cold airmass entering from the south. We can show that the cold airmass is able to reach ATTO, but is not associated with high O3 anymore. The opposite is true its depleted from O3. In the manuscript we give the following explanation: "However, it should be noted that this mass of air rich in O3 did not reach the Manaus region and the ATTO-site. It is believed that the presence of the cloud cover in central Amazonia on 11th, July (Fig. 5), formed by the convergence of air (Friagem and Eastern winds), has an inhibitory effect on O3 formation (Betts et al., 2002). As O3 deposition prevails, a net loss of ozone is expected during transport under conditions of limited photochemical production. The rain forest canopy is a strong sink for ozone (Jacob and Wofsy, 1990; Fan et al., 1990; Rummel et al., 2007). Therefore, the low O3 mixing ratio in the Manaus region and the ATTO-site during the 11th July (Fig. 6-f) would be associated with cloudiness and prolonged transport over forested regions". Flux measurements above amazon rainforests give
consistently high deposition velocities of about 2 cm s-1 around noon (Fan et al., 1990; Rummel et al., 2007). Taking the a simple approach of deriving a lifetime of ozone with respect to deposition, i.e. deposition velocity divided by boundary layer height (Nguyen et al., 2015) gives for noon time conditions and a BL of 1000 m 13 hours and for 500 m of 7 hours, respectively. (2) Actually on all 3 days when the Friagem event occurred in the Manaus region, clouds were present, as shown in Figure 1 of this document (July 9 and 10) and in Figure 5 of the manuscript (July 11). The presence of such cloudiness reduced incident short-wave radiation and O3 near the surface (Fig 10a and 10b of the manuscript). However, it was during 11th July when shortwave radiation suffered the greatest reduction, and therefore we used that day as a case study. (3) We agree in parts with the reviewer. Fig. 6 of the manuscript and Fig. 4 of this document shows the values of the surface concentration of O3, obtained through ERA5, before (Fig. 6a) and during Friagem (Fig. 6b). It is possible to clearly notice that the Friagem (cold pool) carries high levels of O3 from the southwest to the central region of the Amazon. However, this air mass has its O3 concentration reduced as it approaches the surrounding region of Manaus (ATTO, T2, T3 and T0z). We believe that the cause of this reduction is the presence of strong cloudiness above this region (Fig. 5 of the manuscript), responsible for the reduction of solar radiation reaching the surface (Fig. 10a) and consequently a decrease in O3, as already highlighted in the manuscript (L: 175-184). Furthermore, a cold airmass occupying the lowest 500m of the BL was clearly identified on the 11th. (4) The argumentation is not speculative because if we argue that photochemistry is absent just the transport and deposition terms of the budget equation remain. Furthermore, it has been shown for the Amazon rainforest that at "very low" NOx-levels (rainy season), the O3 budget is controlled by downward transport (i.e. vertical mixing) and deposition to the canopy (Jacob and Wofsy, 1990). Additionally, there is a small photochemical loss (Jacob and Wofsy, 1990). Due to increase cloudiness, this contribution will be also small in our case. For the dry season ("higher-NOx") O3 vales have been found to be mainly controlled by photochemistry and by deposition to the forest (Jacob and Wofsy, 1988). Again consistent with the argumentation, that if
photochemistry is reduced due to increased cloudiness the deposition term will persist and increase loss of O3. The referee is right that we do not provide numbers, but the observed phenomena are consistent with the argumentation. The argumentation that reduced vertical mixing is (at least partly) is responsible for very low O3 values refers to the situation on the 11th as with the largest drop in surface O3 at the same time large accumulation of CO2 (emitted by the forest) was observed. The large CO2 values are difficult to explain by the action of convective systems, but they fit to the reduced O3 values due to reduced vertical mixing (generally convective systems also increase surface O3 by downward transport). Furthermore, for the 11th there evidence from a) the wind field in the BRAMS model (fig 12a in manuscript), b) the potential temperature profiles of the BRAMS (Fig. 3 in this document) and the boundary layer height of just 500 m from ERA5 that there is a colder air mass (cold pool) near the surface (fig 12b in manuscript), that traps trace gases close to the surface.

**Question**

The general features of the cold front are clearly described in the manuscript such as the temperature drops and the trade wind is weakened, which accounts for the majority of the manuscript. However, the understanding and discussion of its mechanism is lacking. For example, (1) it is not clear how the cold front induces the convection on July 11 that affects the O3, and thus it's still unclear to what extent Friagem affects O3 in general without knowing its influences on inducing convections. (2) In addition, the cold pool and the subsequent weakened vertical mixing are not well demonstrated because of the lack of vertical profiles of meteorological variables. I believe these can be fixed by further exploring the model results.

**Answer**

(1) We believe that the arrival of Friagem in the central region of the Amazon (region around Manaus and the ATTO site) brings with it a layer of cold, dry air that meets the hot and humid air coming from the Eastern Amazon region (L152-158 of the old
vertion of manuscript). This will favor the formation of convective clouds in this region. Marengo et al. (1997) draw attention to this effect (page 1565): "Based on the observations of wind speed and direction and cloudiness, along with the air temperature data, it is suggested that cold-air advection is the main mechanism for cooling in Ji-Paraná where maximum and minimum air temperatures fell substantially and the sky remained cloud free. At Marabá and Manaus increased cloudiness (probably middle-level clouds or shallow cumulus), associated with the colds, meant that the cooling took the form of reduced maximum temperatures and reduced diurnal temperature range." The satellite images (Fig. 5 of the manuscript) show the presence of clouds during the arrival of the Friagem at the ATTO site. With the help of the BRAMS simulations we will explain the formation of these clouds a little better. Fig. 3a of this document shows the divergence of the horizontal wind obtained by the reanalysis of Era5 on July 11th at 12UTC, where there is also a red square demarcating the area of the domain used in the simulation with the JULES-CCATT-BRAMS model. There is a band of convergence of the westerly and easterly winds, passing through the region of the ATTO site, where convective activity was also formed, as seen in Figure 5 of the manuscript. Fig. 3b shows the distribution of precipitation and the horizontal wind at 15:30 UTC on the 11th of July (simulated with the JULES-CCATT-BRAMS model). These results make it possible to visualize the circulation of the Lake Balbina breeze and some storms formed nearby of the ATTO site. In addition, even though the domain of the grid used in the simulation is much smaller than the area studied with the reanalysis, it is possible to observe the formation of the storms in the convergence of the southwesterly wind with easterly wind in the same way that was observed in Figure 3a. Fig. 3c (cross-section - line AB in Figure 3b) shows the behavior of current lines u, w together with rain water mix ratio. In the layer from the surface to the level of 1000 meters, the westerly flow converges with the easterly flow in the region where the mature convection is located.

We know that in the presence of solar radiation, volatile organic compounds (VOCs) and nitrogen dioxides (NO + NO2 = NOx), O3 is photochemically produced (Davidson, 1993; Wakamatsu et al. 1996; Gerken et al., 2016). Therefore, the presence of a large
cloud cover in the central region of the Amazon, during the Friagem, reduced the arrival of solar radiation on the surface and consequently the surface concentrations of ozone (Fig. 11 of the manuscript).

(2) We believe that the West-Northwest and Southerly winds at low levels (up to approximately 500 m) and a boundary layer that did not exceeded 500 m (Fig. 12 of the manuscript) are already strong indications of the presence of a cold pool during the occurrence of Friagem. However, we are presenting Fig. 3 that shows the potential temperature profile simulated by JULES-CCATT-BRAMS for the ATTO site. It is possible to notice that in the afternoon of July 11th (the moment when the Friagem was most intense in the region) the potential temperature of the air layer located between the surface up to approximately 500 m is lower than the temperature of the layer immediately above (residual layer). That means that the presence of the cold pool was well captured by the BRAMS model.

**Major comments**

Question

1. Line 145: Figure 3 suggest that the changes in temperature are not that significant for Manaus and ATTO, somewhere within 2 degrees.

**Answer**

We agree with the reviewer that in Figure 3, where reanalysis data are shown, it is not possible to observe significant drops in temperature in the region of Manaus and ATTO (around 2 °C), compared to the drop experienced in Porto Velho (around 6° C). We will rewrite the sentence in the new version of the manuscript (L155-157). However, in Figure 7 of the manuscript, the air temperature values measured experimentally in Manaus and at the ATTO site are shown, where it is noted that the decrease in air temperature was in also of the order of 4 °C during the Friagem event.

**Question**

ACPD
2. Figure 4: Is the same data in Fig. 4 as in Fig. 3a and 3f? I wouldn't show the same data twice.

**Answer**

Figures 3 and 4 were merged into one (in the new version of the manuscript Fig.3)

**Question**

3. Line 160 "carries air rich in O3": What is (are) the source(s) of the O3?

**Answer**

We believe that Friagem carries O3 from the Southeastern region of the Brazil (very polluted) towards the Amazon region, as shown in Figure 4, through reanalysis data. We added a small comment on the manuscript (L172).

**Question**

4. Line 166-167: The chemical reactions with terpenes emitted by the forest might be important for O3 loss too. The estimate of the lifetime of the O3, which is a function of dry deposition and chemical reactions is needed for this argument "As O3 deposition prevails, a net loss of ozone is expected during transport under conditions of limited photochemical production".

**Answer**

The loss by chemical reactions with terpenes in the BL above the amazon rainforest has not yet been directly quantified (to our knowledge). The deposition velocities given are the net deposition and therefore not only consider dry deposition, but also within canopy chemical reactions (including terpenes). From own calculations (unpublished) and also literature (e.g. (Freire et al., 2017)) the contribution of terpenes for this layer is negligible. As Fluxes (from which deposition velocities were derived) were measured shortly above canopy ( $\sim$  40 m above ground level) the chemical reactions are just considered for the volume below this height. The loss of O3 by these reactions considering
the whole mixed layer is therefore uncertain. One can argue that these compounds are emitted by the forest and therefore concentrations at ground level are highest and their contribution to O3 loss diminishes with height. Therefore, the above given estimates of the lifetime with respect to deposition should serve as qualified guess of the total loss rate.

**Question**

5. Line 178-179: How the maximum air temperature is defined here? Seems like it is part of the diurnal cycles, which to me is not an appropriate metric for evaluating the intensity of the Friagem.

**Answer**

We would like to thank the reviewer for the opportunity to better clarify the role of Friagem on air temperature. Agreeing with the reviewer that the difference between maximum and minimum temperature is not an appropriate metric to define the temperature drop produced by a Friagem event. However, we would not like to associate intensity of the Friagem with a drop in temperature, as we believe that such intensity would be associated with several other parameters. We will rewrite the sentence as follows:

"At Porto Velho the difference between the maximum mean air temperature (maximum average daily cycle value) and the maximum air temperature during the Friagem (July 8th) was 7 °C (from 31 to 24°C), whilst in Manaus region and at ATTO the differences were in the order of 4 °C (from 30 to 26 °C and 29 to 25 °C, respectively) during July 11th." (L188-191)

Question

6. Line 213-215: not clear. Clarify.

Answer
During the occurrence of the forest breeze towards Lake Balbina it would be expected that the wind direction would be from East-Southeast, and not from West or North, as noted in Fig. 9 of the manuscript.

We will added a short comment to the sentence clarify this (L230).

Question

7. Line 228: Any explanations for the decreases in O3?

Answer

We believe that we have already answered this question in this document. In summary, we answered that the presence of heavy cloudiness around 13 LT (where maximum O3 concentrations are expected) reduced the incident solar radiation (Fig. 10a) and therefore photochemical production of O3.

**Question**

8. Line 238-241: "did not result in an increase of near surface O3". I don't necessarily agree with this. I think there is an increase in O3 from roughly 6 ppbv to 10 ppbv. To validate if this increase is due to the convection, you can calculate the virtual potential temperature as in Gerken et al. (2016).

**Answer**

In the work of Gerken et al. (2016) the virtual potential temperature was not calculated, but equivalent potential temperature ( $\theta$ e). However, Dias-Júnior et al. (2017), used data from Manacapurú (T3, central Amazon) and showed that the correlation between the  $\theta$ e drop is not well correlated with the superficial increases in O3 (Fig. 6 by Dias-Júnior et al. (2017)), during the occurrence of downdrafts. Also according to Dias-Júnior et al. (2017) a parameter that best represents the superficial increases in O3 is a  $\Delta$ CAPE (difference between the CAPE values immediately before the downdraft and the value after the downdraft). Unfortunately, we do not have data to enable us to
calculate CAPE for the period investigated in this work for ATTO site.

**Question**

9. Line 247 and 262: The vertical mixing can be evaluated by the vertical profiles of the virtual potential temperature.

**Answer**

We do not have temperature profiles for the data period used in this work. Figure 5 shows the virtual potential temperature profiles obtained from JULES-CCATT-BRAMS simulation. On 11th July the virtual potential temperature of the air layer located between the surface up to approximately 500 m is lower than the temperature of the layer immediately above (similar to that shown in Fig. 3), that is, the vertical mixing will be reduced in the presence of Friagen events.

**Question**

10. Figure 13: Why the temperature at 24.4 m is used? It is within the canopy if I understand correctly, which I think would be very different (presumably lower) from above-canopy temperature.

**Answer**

Thank you very much for the comments. The simulated figures at the height of 24.4 m were replaced by the simulated figures at the height of 76.8 m (Fig. 6 in this document).

**Question**

11. How well the surface layer is represented by the JULES-CCATT-BRAMS model in general? How about in this study? Any comparisons between the modelled and the observations to evaluate the fidelity of the model for surface layer?

**Answer**

The formulations of the JULES surface scheme include dynamic vegetation, photo-
synthesis and plant respiration, carbon storage and soil moisture. The JULES surface scheme has been coupled to the CCATT-BRAMS modeling system using an explicit scheme. This coupling is two-way in the sense that, for each model time step, the atmospheric component provides to JULES the current near-surface wind speed, air temperature, pressure, condensed water and downward radiation fluxes, water vapor and trace gas mixing ratios. After its processing, JULES advances its state variables over the time step and feeds back to the atmospheric component the sensible and latent heat and momentum surface fluxes, upward short-wave and long-wave radiation fluxes, as well as a set of trace gas fluxes (Moreira et al, 2013).

Figures 7a-b show the values of the sensible (H) and latent (LE) heat obtained through experimental data above the ATTO site (80 m) and through the JULES-CCATT-BRAMS simulation, respectively (76.8 m). It is possible to notice that the simulation overestimates the values of both flows. However, it is noted that the LE values are higher than the H values, mainly for the daytime period. This result is expected for a forested surface, such as the Amazon rainforest.

**Question**

12. Line 318-319: The suppressed vertical mixing might play a part in the decreased O3 mixing ratios, but it's not the only or main reason here.

**Answer**

As outlined above there is evidence from several sources that the lowest 500 m are occupied by a colder air mass and therefore vertical mixing is suppressed on the 11th. In parallel to reduced O3 mixing ratios we observed accumulation of CO2 which gives further evidence for trapping of trace gases in this layer. In absence of considerable photochemical activity, the situation can be seen as similar to the nocturnal boundary layer where consistently (vast body of literature) loss of O3 by deposition and chemical reactions is observed and increases in concentration are due to intermittent vertical mixing esp. by occurrence of low level jets. Therefore, we think that the reduced
vertical mixing has a strong influence on the near surface values, but to clarify that it might not be the sole reason we now write that it "contributes" to the reduced values. (L334).

Minor comments

Question

1. Line 66: I'd cite more relevant studies regarding O3 at the T3 site.

Answer

Was done. Thank you.

Question

2. Line 67: I'd point out the minimal anthropogenic influences at the ZF2 site to contrast the other sites.

Answer

Was done. Thank you.

Question

3. Table 1: What is the canopy height at ATTO site?

Answer

The average height of trees at ATTO site is approximately 37 m (Andreae et al., 2015).

Question

4. Figure 7: I'd present the data in the order of Porto Velho, Manaus, and ATTO.

Answer

Was done.
Question

5. Line 207: There are some editorial/technical issues to be fixed. For example, the parentheses are missing for "Fig. 9".

Answer

Was corrected. Thank you.

References

Andreae, M. O., et al.: The Amazon Tall Tower Observatory (ATTO): overview of pilot measurements on ecosystem ecology, meteorology, trace gases, and aerosols, Atmos. Chem. Phys., 15, 10 723–10 776, https://doi.org/10.5194/acp-15-10723-2015, 2015.

Davidson, A., 1993, Update on ozone trend in California's south coast air basin. Journal of Air and Waste Management Association, 43, pp. 226–227.

Dias-Júnior, C. Q., Dias, N. L., Fuentes, J. D., and Chamecki, M.: Convective storms and non-classical low-level jets during high ozone level episodes in the Amazon region: An ARM/GOAMAZON case study, Atmos. Environ., 155, 199–209, https://doi.org/https://doi.org/10.1016/j.atmosenv.2017.02.006, 2017.

Fan, S.-M., Wofsy, S. C., Bakwin, P. S., Jacob, D. J., and Fitzjarrald, D. R.: Atmospherebiosphere exchange of CO2 and O3 in the central Amazon forest, J. of Geophys. Res.-Atmos., 95, 16 851–16 864, https://doi.org/10.1029/JD095iD10p16851, 1990.

Freire, L. S., Gerken, T., Ruiz-Plancarte, J., Wei, D., Fuentes, J. D., Katul, G. G., Dias, N. L., Acevedo, O. C., and Chamecki, M. Turbulent mixing and removal of ozone within an Amazon rainforest canopy, J. Geophys. Res. Atmos., 122, 2791–2811, doi:10.1002/2016JD026009, 2017.

Gerken, T., Wei, D., Chase, R. J., Fuentes, J. D., Schumacher, C., Machado, L. A., ... & Jardine, A. B. (2016). Downward transport of ozone rich air and implications for atmospheric chemistry in the Amazon rainforest. Atmospheric Environment, 124,
64-76.

Jacob, D. J. and Wofsy, S. C.: Budgets of reactive nitrogen, hydrocarbons, and ozone over the Amazon forest during the wet season, J. Geophys. Res., doi:10.1029/jd095id10p16737, 1990.

Marengo, J. A., Nobre, C. A., and Culf, A. D.: Climatic impacts of "friagens" in forested and deforested areas of the Amazon basin, J. Appl. Meteorol., 36, 1553–1566, https://doi.org/10.1175/1520-0450(1997)036<1553:CIOFIF>2.0.CO;2, 1997.

Moreira, D. S., Freitas, S. R., Bonatti, J. P., Mercado, L. M., Rosário, N. M. E., Longo, K. M., ... & Gatti, L. V. (2013). Coupling between the JULES land-surface scheme and the CCATT-BRAMS atmospheric chemistry model (JULES-CCATT-BRAMS1. 0): applications to numerical weather forecasting and the CO2 budget in South America. Geoscientific Model Development, 6(4), 1243-1259.

Rummel, U., Ammann, C., Kirkman, G., Moura, M., Foken, T., Andreae, M., and Meixner, F.: Seasonal variation of ozone deposition to a tropical rain forest in southwest Amazonia, Atmos. Chem. Phys., 7, 5415–5435, https://doi.org/10.5194/acp-7-5415-2007, 2007.

Wakamatsu, S., OHARA, T. and UNO, I., 1996, Recent trends in precursor concentrations and oxidant distribution in the Tokyo and Osaka areas. Atmospheric Environment, 30, pp. 715–721.

Please also note the supplement to this comment: https://acp.copernicus.org/preprints/acp-2020-564/acp-2020-564-AC2-supplement.pdf
Figure 1: Enhanced images of the GOES 13 satellite in the infrared channel on: (a) July  $09^{th}$  at 17:00 UTC and (b) July  $10^{th}$  at: 17:30 UTC.

Fig. 1.

---

## Author Response (AR2)

*Editor Decision: Publish subject to minor revisions (review by editor) (09 Nov 2020) by Gilberto Fisch*

*Comments to the Author:*

*The authors did most of the points raised by the 2 reviewers. However, as the reviewer B point out, the question of the drop of air temperature was not clearly addressed. Moreover, I would like to see if the specific humidity (or better the equivalent potential temperature) will show a difference along the days...This is to exclude the possibility of wind direction/O3 enrich by convection (General Comments (1) and Major Comments (1) and (5)). There are negative values of Heat Fluxes that can be useful for your analysis, explore Fisch (1996) his PhD Thesis. I also recommend to read/incorporate 2 previous study about Friagem in Amazonia, Viana and Herdies (2018) DOI http://dx.doi.org/10.1590/0102-7786331014 and Ricarte et al. (2014) https://doi.org/10.1002/met.1458.*

We would like to thank the Editor for his comments/suggestions. We incorporated the references (Fisch, 1996; Ricarte et al., 2014; Viana and Herdies, 2018) cited by the Editor in the new version of the manuscript (in blue). We believe that these works bring important contributions to a better understanding of the role of Friagem phenomena in the thermodynamics of the atmosphere.

We will start by showing the values of the sensible heat flux ($H$) obtained with experimental data before and during the arrival of the Friagem at the ATTO site (Figure 1). We emphasize that it was on July 11, 2014 that Friagem arrived in the central region of the Amazon. It is noted that the $H$ values did not show strong decreases during the arrival of the Friagem compared to the $H$ values in the days before the Friagem. It is also observed that in some times the values of $H$ decrease sharply (in some cases these values become negative even in daytime conditions). These falls occurred during the presence of deep convection above the region (Figure 2).

Figure 3a shows the values of air temperature and equivalent potential temperature ($\theta_e$) and Figure 3b shows the values of specific humidity ($q$) measured at the ATTO experimental site. There is a slight drop in air temperature with the arrival of Friagem, as mentioned in the Manuscript (L190). The values of $q$ during the Friagem practically did not vary compared to the values observed in the previous days. Sharp drops in the values of $q$ and $\theta_e$ are observed, probably much more associated with the presence of downdrafts from convective clouds (Betts et al., 2002; Gerken et al., 2016; Dias-Junior et al., 2017), than with the arrival of the Friagem.

The works indicated by the Editor (Fisch, 1996; Ricarte et al., 2014; Viana and Herdies, 2018) and others (Marengo et al., 1997; Silva-Dias et al., 2004) show that the arrival of Friagem in the South region of Amazônia produced striking changes in the ABL thermodynamic variables. For example, Fisch (1996) showed that the values of $H$ are considerably lower during the presence of Friagens (page 140, Fig. A4-a). Ricarte et al. (2014) showed that both maximum and minimum temperatures suffer strong reductions with the arrival of Friagem. Viana and Herdies (2018) also noticed sharp drops in temperature and specific air humidity in the South of the Amazon region. However, they noted that in the central Amazon region (Manaus) these reductions were quite weak, similar to the results shown in the manuscript and here. According to Marengo et al. (1997) and Silva-Dias et al (2004) one of the main effects of Friagem in the central region of the Amazon is the induction of strong cloudiness. We believe that the presence of these clouds reduces the solar radiation that reaches the surface and plays an important role in the surface concentration of O3, as mentioned in the manuscript (L181-184; L329-335).

[Figure]

Figure1: Sensible heat flux on July 07-11th 2014, measured at 81 m, at ATTO site.

[Figure]

Figure 2: Enhanced images of the GOES 13 satellite in the infrared channel on: (a) July 09[th] at 17:00 UTC, (b) July 10[th] at: 17:30 UTC and (c) July 11[th] at: 16:00 UTC.

[Figure]

Figure3: (a) Air temperature and equivalent potential temperature ($\theta_e$). (b) Specific humidity on July 07-11[th] 2014, measured at 81 m, at ATTO site.

**References**

Betts, A. K., Gatti, L. V., Cordova, A. M., Dias, M. A. S., and Fuentes, J. D.: Transport of ozone to the surface by convective downdrafts at night, J. of Geophys. Res.-Atmos., 107, LBA–13, 2002.

Dias-Júnior, C. Q., Dias, N. L., Fuentes, J. D., and Chamecki, M.: Convective storms and non-classical low-level jets during high ozone level episodes in the Amazon region: An ARM/GOAMAZON case study, Atmos. Environ., 155, 199–209, https://doi.org/https://doi.org/10.1016/j.atmosenv.2017.02.006, 2017.

Fisch, G.F. Camada Limite Amazônica: Aspectos Observacionais e de Modelagem. 180 f. Tese (Doutorado em Meteorologia) - Instituto Nacional de Pesquisas Espaciais, São José dos Campos, 1996.

Gerken, T., Wei, D., Chase, R. J., Fuentes, J. D., Schumacher, C., Machado, L. A., Andreoli, R. V., Chamecki, M., de Souza, R. A. F., Freire, L. S., Jardine, A. B., Manzi, A. O., dos Santos, R. M. N., von Randow, 435 C., dos Santos Costa, P., Stoy, P. C., Tóta, J., and Trowbridge, A. M.: Downward transport of ozone rich air and implications for atmospheric chemistry in the Amazon rainforest, Atmos. Environ., 124, Part A, 64 – 76, https://doi.org/10.1016/j.atmosenv.2015.11.014, 2016.

Marengo, J. A., Nobre, C. A., and Culf, A. D.: Climatic impacts of "friagens" in forested and deforested areas of the Amazon basin, J. Appl. Meteorol., 36, 1553–1566, https://doi.org/10.1175/1520-0450(1997)036<1553:CIOFIF>2.0.CO;2, 1997.

Ricarte RM, Herdies DL, Barbosa TF. Patterns of atmospheric circulation associated with cold outbreaks in southern Amazonia. Meteorological Applications. 2015 Apr;22(2):129-40.

Silva Dias, M., Dias, P. S., Longo, M., Fitzjarrald, D. R., and Denning, A. S.: River breeze circulation in eastern Amazonia: observations and modelling results, Theor. Appl. Climatol., 78, 111–121, https://doi.org/10.1007/s00704-004-0047-6, 2004

Viana LP, Herdies DL. Estudo de caso de um evento extremo de incursão de ar frio em julho de 2013 sobre a bacia amazônica brasileira. Revista Brasileira de Meteorologia. 2018 Mar;33(1):27-39.